# Textual Unlearning Gives a False Sense of Unlearning

**Jiacheng Du** [1 2]  **Zhibo Wang**[† 1 2]  **Jie Zhang** [3]  **Xiaoyi Pang** [1 2]  **Jiahui Hu** [1 2]  **Kui Ren** [1 2]

## Abstract

Language Models (LMs) are prone to "memorizing" training data, including substantial sensitive user information. To mitigate privacy risks and safeguard the right to be forgotten, machine unlearning has emerged as a promising approach for enabling LMs to efficiently "forget" specific texts. However, despite the good intentions, is textual unlearning really as effective and reliable as expected? To address the concern, we first propose Unlearning Likelihood Ratio Attack+ (U-LiRA+), a rigorous textual unlearning auditing method, and find that unlearned texts can still be detected with very high confidence after unlearning. Further, we conduct an in-depth investigation on the privacy risks of textual unlearning mechanisms in deployment and present the Textual Unlearning Leakage Attack (TULA), along with its variants in both black- and white-box scenarios. We show that textual unlearning mechanisms could instead reveal more about the unlearned texts, exposing them to significant membership inference and data reconstruction risks. Our findings highlight that existing textual unlearning actually gives a false sense of unlearning, underscoring the need for more robust and secure unlearning mechanisms.

## 1. Introduction

In recent years, language models (LMs) have demonstrated impressive capabilities, driven by extensive training samples. Among them, a significant portion of real-world user data is contained to improve LMs in deployment. For instance, Amazon collects user conversations via smart speakers to enhance command-interaction capabilities (Valinsky, 2019), and OpenAI gathers user inputs to improve Chat-

[1]The State Key Laboratory of Blockchain and Data Security, Zhejiang University, P. R. China [2]School of Cyber Science and Technology, Zhejiang University, P. R. China [3]ETH Zurich, Switzerland. Correspondence to: Zhibo Wang <zhibowang@zju.edu.cn>.

*Proceedings of the 42nd International Conference on Machine Learning*, Vancouver, Canada. PMLR 267, 2025. Copyright 2025 by the author(s).

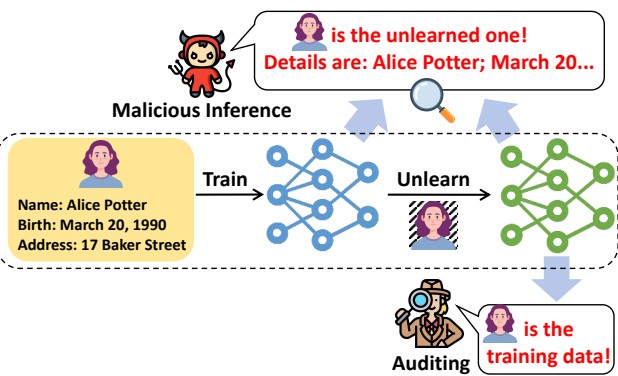

Figure 1. Textual unlearning gives a false sense of unlearning. Our findings highlight that: (1) the unlearned data could still be detected after unlearning via rigorous auditing. (2) An adversary could instead infer the unlearned data by analyzing the models before and after unlearning.

GPT (O'Flaherty, 2024). However, user data often contains sensitive information, such as phone numbers and home addresses (Liu et al., 2023). Recent studies have revealed that the training data of LMs can be maliciously inferred or extracted (Mattern et al., 2023; Carlini et al., 2021), raising serious privacy concerns. Besides, the users are entitled to the "Right To Be Forgotten" (Rosen, 2011). Various regulations, such as the General Data Protection Regulation (GDPR, 2018), grant users the right to request a complete deletion of personal data, even when it has already been trained. Addressing these concerns necessitates the mechanisms to "undo" the impact of specific training samples on LMs, safeguarding data security and individual rights.

Machine unlearning (MU) aims to erase the impact of specific training samples on a trained model (Bourtoule et al., 2021). There are generally two categories: *exact unlearning* and *inexact unlearning*. *Exact unlearning* directly removes the unlearned samples from the training set and retrains the model on the remaining samples (Bourtoule et al., 2021). While straightforward, this approach is computationally expensive, particularly for LMs with extensive training samples and parameters. Therefore, *inexact unlearning* methods have attracted much attention, aiming to efficiently approximate the retrained model without retraining from scratch. They typically fine-tune the original model with the unlearned samples and an "inverse learning" objective, such as gradient ascent (Jang et al., 2022). After a few fine-tuning

steps, the unlearned model behaves as if it were a retrained model that has not been trained on unlearned data.

Despite promising intentions, *existing MU methods may be less effective than expected*. Recent studies suggest that the traces of the unlearned data, such as membership (Hayes et al., 2024) or adversarial image triggers (Pawelczyk et al., 2024), may not be completely erased after unlearning. Besides, MU methods may reveal extra information about unlearned data. Specifically, comparing models before and after unlearning could further expose the membership (Chen et al., 2021) or visual features (Hu et al., 2024) of the unlearned samples. However, related studies primarily focus on image data, while the vulnerability and privacy risks of textual unlearning on LMs remain under-explored.

To fill the gap, we conduct a deep investigation into textual unlearning on LMs. From the first principle, a successful textual unlearning process should at least ensure that *the memory of unlearned texts is completely erased* from the model and *no new privacy risks are further introduced*. Thus, we propose two key research questions (RQ):

● **RQ1:** Can textual unlearning really achieve clean erasure?

● **RQ2:** Will textual unlearning backfire and pose new privacy risks?

To address **RQ1**, we introduce a *rigorous auditing method* called *Unlearning Likelihood Ratio Attack+* (U-LiRA+). By auditing the *most-vulnerable* samples in the training set (Aerni et al., 2024), U-LiRA+ enables a rigorous evaluation on the erasing effectiveness for existing unlearning methods in the worst case.

For **RQ2**, we explore the privacy risks of textual unlearning mechanisms in deployment by developing *Textual Unlearning Leakage Attacks* (TULA). In the black-box scenarios, we employ *TULA for Membership Inference* (TULA-MI), which enables an adversary to infer the membership of the unlearned text by querying the models before and after unlearning. In the white-box scenarios, we propose *TULA for Data Reconstruction* (TULA-DR). With access to model weights, the adversary could effectively reconstruct the unlearned texts via continuous constrained optimization.

Through extensive evaluations, our findings critically reveal that **textual unlearning actually gives a false sense of unlearning!** Existing methods for textual unlearning *fail to completely erase unlearned texts*, and their deployment will *instead introduce heightened privacy risks*. Specifically, we highlight the following key findings:

- **Unlearned texts remain detectable with high confidence on the unlearned LMs**. We argue that previous unlearning auditing methods highly overestimate the erasing effectiveness of existing textual unlearning methods. In the worst case, over 70% of unlearned

texts can still be correctly inferred after unlearning, with an error rate below 0.1%.

- **The textual unlearning mechanism additionally introduces new risks of leaking membership information on the unlearned texts**. By querying the model before and after unlearning, a malicious user can confidently infer the membership of unlearned texts by comparing the model outputs, even when only loss values are available.

- **The textual unlearning mechanism poses a substantial risk of malicious reconstruction on unlearned texts**. With access to model weights before and after unlearning, an adversary could reconstruct unlearned texts with even more than 80% accuracy.

Our approaches, along with the concurrent works (Pawelczyk et al., 2024; Hayes et al., 2024; Huang et al., 2024; Shumailov et al., 2024; Hu et al., 2024), highlight the importance of rigorous evaluation for unlearning mechanisms. Prior to widespread deployment, these mechanisms should undergo thorough auditing and analysis to prevent misleading conclusions and potential security risks.

## 2. Related Works and Preliminaries

### 2.1. Machine Unlearning

Machine unlearning aims to erase the impact of specific training data on a trained model (Bourtoule et al., 2021). There are generally two categories: *exact unlearning* and *inexact unlearning*. *Exact unlearning* methods remove the unlearned samples ($\mathcal{D}_{unlearn}$) from the training set ($\mathcal{D}$) and retrain the original model ($\mathcal{M}_{original}$) on the remaining data ($\mathcal{D}_{remian}$) from scratch, which provides the cleanest erase and is considered as the "gold standard" (Bourtoule et al., 2021). However, *exact unlearning* has a high computational overhead in practice, especially for LMs, given their large training corpus and massive parameters. To improve practicality and efficiency, *inexact unlearning* methods are proposed to efficiently approximate the retrained model without retraining from scratch. Their basic idea is to fine-tune $\mathcal{M}_{original}$ on the $\mathcal{D}_{unlearn}$ with an "inverse learning" objective. Specifically, Jang et al. propose to fine-tune $\mathcal{M}_{original}$ by Gradient Ascent (GA) based on an intuition that unlearning should be the inverse process of learning (Jang et al., 2022). Later, Chen et al. propose to maximize the Kullback-Leibler divergence of the outputs between $\mathcal{M}_{original}$ and unlearned model ($\mathcal{M}_{unlearn}$) on $\mathcal{D}_{unlearn}$ (Chen & Yang, 2023). Besides, Zhang et al. treat $\mathcal{D}_{unlearn}$ as negative preference samples and transform unlearning to a Negative Preference Optimization (NPO, (Rafailov et al., 2024)) problem on LMs (Zhang et al., 2024b).

## 2.2. Machine Unlearning Auditing

A successful unlearning process should ensure that the $\mathcal{M}_{unlearn}$ does not disclose any membership information, indicating that $\mathcal{D}_{unlearn}$ is part of $\mathcal{D}$. Therefore, Membership Inference Attacks (MIAs) (Shokri et al., 2017) are widely used to audit MU methods (U-MIAs). MIAs aim to determine whether a sample belongs to the training set by analyzing model output discrepancies (e.g., loss values). Consequently, if the adversary cannot confidently distinguish whether a sample originates from the $\mathcal{D}_{unlearn}$ or the unseen dataset $\mathcal{D}_{unseen}$ (e.g., nearly random guess) on $\mathcal{M}_{unlearn}$, the unlearning process is deemed successful. Existing MU methods mainly use population-based U-MIAs for auditing (Shi et al., 2024), and recent works (Hayes et al., 2024; Pawelczyk et al., 2024) attempt to use per-sample U-MIAs to improve auditing precision.

**Population-based U-MIAs.** Previous studies typically employ simple population-based MIAs to audit MU methods. The general approach involves an adversary selecting $K/2$ samples from $\mathcal{D}$ and unlearning them with the audited MU method to obtain $\mathcal{M}_{unlearn}$. Subsequently, $K/2$ unseen samples (e.g., holdout data from $\mathcal{D}$) are randomly chosen to form an auditing set of size $K$. The adversary then guesses that one of the audited samples $x$ belongs to $\mathcal{D}$ if $\mathcal{S}(\mathcal{M}_{unlearn}; x) > \tau$, where $\mathcal{S}$ is a scoring function (e.g., negative cross-entropy loss) and $\tau$ is a threshold. If the attack accuracy, typically measured using AUC-ROC (Shokri et al., 2017), is close to 50%, it indicates that the adversary cannot distinguish $\mathcal{D}_{unlearn}$ and $\mathcal{D}_{unseen}$ on $\mathcal{M}$, suggesting that the unlearning process has been successful.

**Per-sample U-MIAs.** Per-sample U-MIA (Hayes et al., 2024; Pawelczyk et al., 2024) builds upon the Likelihood Ratio Attack (LiRA) method proposed by Carlini et al. (Carlini et al., 2022), which formulates MIA as a hypothesis testing problem. For an audited sample $x$, the adversary models the score distributions under the hypotheses that $x$ is a member of $\mathcal{D}_{unlearn}$, and that $x$ is one of $\mathcal{D}_{unseen}$. Given the score of $x$ on $\mathcal{M}_{unlearn}$, the attack then applies *a likelihood ratio* test to distinguish between the two hypotheses. To estimate the distributions, the adversary trains multiple shadow models (Shokri et al., 2017) by repeatedly sampling an identically distributed auxiliary training set $\mathcal{D}'$. For the target $x$, the adversary trains the unseen shadow model $\mathcal{M}_{out}$ on $\mathcal{D}'$, and trains the unlearn shadow model $\mathcal{M}_{in}$ on $x \cup \mathcal{D}'$ after unlearning $x$. With a sufficient number of shadow models, the adversary fits two Gaussians $\mathcal{N}(\mu_{x,\text{in}}, \sigma_{x,\text{in}}^2)$ and $\mathcal{N}(\mu_{x,\text{out}}, \sigma_{x,\text{out}}^2)$ to the scores from $\mathcal{M}_{in}$ and $\mathcal{M}_{out}$ models on the target sample $x$. Finally, the adversary applies a standard Neyman–Pearson test to determine whether the observed score from the $\mathcal{M}_{unlearn}$

is more likely if $x$ is an unlearned or unseen sample:

$$\mathcal{A}(\mathcal{M}_{unlearn}, x) := \frac{\mathcal{N}(\mathcal{S}(\mathcal{M}_{unlearn}; x) \mid \mu_{x,\text{in}}, \sigma_{x,\text{in}}^2)}{\mathcal{N}(\mathcal{S}(\mathcal{M}_{unlearn}; x) \mid \mu_{x,\text{out}}, \sigma_{x,\text{out}}^2)}.$$

Overall, per-sample U-MIAs provide more precise auditing than population-based methods, since the methods discriminate each sample independently rather than with a global threshold. However, we argue that all the existing auditing methods for unlearning are not rigorous, as they fundamentally fail to properly select the audited samples.

## 2.3. Privacy Attacks against Machine Unlearning

Existing privacy attacks on MU generally follow a basic idea: inferring the privacy of the unlearned data via comparing the models before and after the unlearning process. Based on the adversary's objective, there are two primary paradigms: MIA and Data Reconstruction Attack (DRA).

Chen et al. demonstrate that the MU mechanism can inadvertently reveal *membership information* about unlearned data (Chen et al., 2021). Specifically, for a target sample $x$, an adversary randomly selects an identically distributed auxiliary dataset $\mathcal{D}'$ and trains two models, $\mathcal{M}'_{original}$ and $\mathcal{M}'_{unlearn}$ (after unlearning $x$), on $x \cup \mathcal{D}'$. The adversary queries both models with $x$, combining their outputs to create the positive feature. For the negative feature, the adversary randomly selects another unseen sample, queries both models, and combines the outputs. By repeating this process multiple times, a dataset of feature pairs is constructed to train a binary classification attack model determining whether $x$ is a member of training data. Hu et al. demonstrate that adversaries can *reconstruct an unlearned sample* by exploiting the difference in model weights before and after unlearning, $\Delta \theta = \mathcal{M}_{original} - \mathcal{M}_{unlearn}$ (Hu et al., 2024). This approach relies on the assumption that $\nabla^*$ aligns with the gradient direction of the unlearned sample $x$ on $\mathcal{M}_{original}$. Specifically, the adversary begins by inferring the label $y'$ of the unlearned sample using a probing technique. Next, the adversary randomly initializes a dummy sample $x'$ with the same shape as $x$ and computes the gradient $\nabla'$ of $(x', y')$ on $\mathcal{M}_{original}$. By minimizing the cosine similarity between $\Delta \theta$ and $\nabla'$, the adversary iteratively updates $x'$ until convergence. However, existing studies focus solely on image data, and the potential privacy threats against textual unlearning remain underexplored.

## 3. Rigorous Auditing on Textual Unlearning

Let's go back to the drawing board to rethink the unlearning auditing. The basic idea is to test whether an adversary could confidently distinguish an unlearned sample from an unseen sample through outputs on $\mathcal{M}_{unlearn}$. However, existing auditing methods ignore a crucial and fundamental

issue: *the proper selection of audited samples.* Typically, these methods randomly partition the dataset into unseen data $\mathcal{D}_{unseen}$ and training data $\mathcal{D}$. A subset of $\mathcal{D}$ is then randomly selected as unlearned data $\mathcal{D}_{unlearn}$, while an equal number of unseen and unlearned samples are designated as the audited samples. However, this approach inadvertently causes $\mathcal{M}_{original}$ to exhibit *"false memory"*, where the model acts as if it had been trained on unseen samples. Such results mainly arise from the model's ability to generalize, as the audited samples are from the same distribution. Furthermore, the inherent overlap and redundancy among text samples exacerbate such an effect (Duan et al., 2024).

The *false memory* effect leads to considerable overlap between the output distributions of $\mathcal{D}_{unlearn}$ and $\mathcal{D}_{unseen}$ on $\mathcal{M}_{original}$, which makes MIA-based auditing methods unable to well distinguish between the samples even before unlearning. This flawed foundation undermines the fairness and rigor of unlearning auditing, as the distinguishability between $\mathcal{D}_{unlearn}$ and $\mathcal{D}_{unseen}$ could be inherently poor.

To conduct rigorous unlearning auditing, it is crucial to utilize audited samples with *"zero memory"*, ensuring *the model knows nothing about them before being trained*. To address this, we propose U-LiRA+, which performs rigorous auditing on unlearning algorithms by additionally constructing and injecting *mislabeled samples*. *Mislabeled samples* naturally occur in real-world datasets. For instance, in a sentiment classification task, a text such as "I love this movie so much!" may be incorrectly labeled as *negative* instead of *positive* due to human error. Since mislabeled samples are typically small in number and counterfactual, a normal model cannot generalize to them unless explicitly trained (Aerni et al., 2024). As a result, mislabeled samples are considered as *the most vulnerable samples* to unlearning, since whether they have been trained makes a significant difference in model outputs. In other words, their presence provides *a worst-case scenario*, enabling fair and rigorous auditing on the effectiveness of unlearning algorithms.

The detailed process of U-LiRA+ is outlined in Algorithm 1. To rigorously audit the unlearning algorithm $U$, we first sample an audit dataset $\mathcal{D}_{audit}$ from the training data distribution and mislabel it. Next, we create a binary mask vector to randomly divide $\mathcal{D}_{audit}$ into two equal parts: unlearned samples and unseen samples. We then train a test model $M^*$ to evaluate the effectiveness of $U$. For each audited sample, we generate $T$ groups of shadow models to estimate the distribution of model outputs when the sample is either unlearned or unseen. Subsequently, we query $M^*$ with the audited sample and determine which distribution its output corresponds to, predicting whether the sample has been unlearned. Finally, we compute the $TPR@LowFPR$ of the predictions against the ground truth mask. If the value is (nearly) zero, the unlearning process is deemed successful.

---

**Algorithm 1** U-LiRA+

**Args:** model $\mathcal{M}$, mislabel function $\mathcal{Q}$, learning algorithm $A$, unlearning algorithm $U$, number of shadow models $T$, size of audited set $N$, logit function $\phi$.

$\mathcal{D}_{audit} \sim D$   sample the audit set
$\mathcal{D}_{audit} \leftarrow \mathcal{Q}(\mathcal{D}_{audit})$   mislabel the audited samples
$Mask \leftarrow \{m_i \mid m_i \sim Uniform(0,1), i = 1, \ldots, N\}$
$\mathcal{D}^* \sim D$   sample a tested training set
$\mathcal{M}^* \leftarrow A(\mathcal{D}^* \cup Mask \odot \mathcal{D}_{audit})$   inject audited samples
$\mathcal{M}^* \leftarrow U(\mathcal{M}^*, Mask \odot \mathcal{D}_{audit})$   unlearn audited samples
$Preds \leftarrow \{\}$
**while** $n \le N$ **do**
  $O \leftarrow \{\}, \hat{O} \leftarrow \{\}$
  **while** $t \le T$ **do**
    $\mathcal{D} \sim \bar{D}$   sample a shadow training dataset
    $\mathcal{D} \leftarrow \mathcal{D} \cup (x_n, y_n^-)$   inject a audited sample
    $\mathcal{M}_{original} \leftarrow A(\mathcal{D})$
    $\mathcal{M}_{unlearn} \leftarrow U(\mathcal{M}_{original}, (x_n, y_n^-)))$
    $\mathcal{M}_{retrain} \leftarrow A(\mathcal{D} \backslash (x_n, y_n^-))$
    $O[t] \leftarrow \phi((x_n, y_n^-); \mathcal{M}_{unlearn})$
    $\hat{O}[t] \leftarrow \phi((x_n, y_n^-); \mathcal{M}_{retrain})$
  **end while**
  $\mu, \sigma \leftarrow$ fit Gaussian$(O)$
  $\hat{\mu}, \hat{\sigma} \leftarrow$ fit Gaussian$(\hat{O})$
  $o \leftarrow \phi((x_n, y_n^-); \mathcal{M}^*)$
  $p_{member} \leftarrow \frac{\mathcal{N}(o;\mu,\sigma^2)}{\mathcal{N}(o;\mu,\sigma^2) + \mathcal{N}(o;\hat{\mu},\hat{\sigma}^2)}$
  **If** $p_{member} > \frac{1}{2}$ **Then** $Preds[n] \leftarrow 1$ **Else** $Preds[n] \leftarrow 0$
**end while**
**Return** $TPR@LowFPR \leftarrow ROC(Preds, Mask)$

---

## 4. Textual Unlearning Leakage Attacks

In this section, we investigate the potential privacy risks of textual unlearning mechanisms in deployment. We propose the Textual Unlearning Leakage Attack (TULA) and its variants in black- and white-box scenarios: TULA for Membership Inference (TULA-MI) and TULA for Data Reconstruction (TULA-DR).

### 4.1. Problem Statement

In an ML system, there are three primary roles: data contributors, the model developer, and beneficiaries (including users and collaborators), as shown in Figure 2. Data contributors provide their private data to the model developer, who utilizes the samples to train a model. Beneficiaries are categorized based on their access to the models. Users can query the model via the developer's API (e.g., ChatGPT (OpenAI, 2023)). In contrast, collaborators, such as organizations contracting with the developer, have full access to the weights and can modify or deploy the model locally.

Suppose a contributor requests the developer to delete its private data $x$. Upon receiving this request, the developer will utilize unlearning algorithms to obtain $\mathcal{M}_{unlearn}$ and remove the unlearned $x$ from the database. Subsequently, both the user-oriented API and the weights shared with collaborators will be updated to completely eliminate the contribution of $x$ to the system. However, the unlearning mechanism

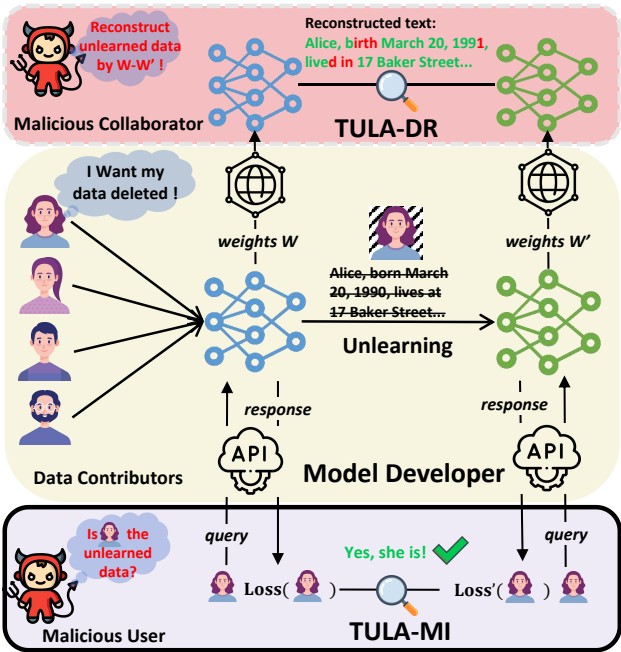

*Figure 2.* An overview of **T**extual **U**nlearning **L**eakage **A**ttack (**TULA**) against ML systems. TULA for Membership Inference (**TULA-MI**) infers the membership of the unlearned texts by querying the models before and after unlearning. TULA for Data Reconstruction (**TULA-DR**) reconstructs the unlearned texts using the weight differences between the two models.

inevitably creates a new risk, even $x$ are perfectly unlearned: users and collaborators can access both the models before ($\mathcal{M}_{original}$) and after unlearning ($\mathcal{M}_{unlearn}$), which *instead emphasizes the "non-existence" of the unlearned sample* and opens a new window for inferring $x$.

### 4.2. TULA for Membership Inference

In this subsection, we introduce TUAL-MI, where a *malicious user* aims to infer the membership information of unlearned data through black-box model access. Depending on different levels of black-box knowledge, we investigate both *strict* and *relaxed* black-box cases to reveal the adversary's threat boundaries in this scenario.

#### 4.2.1. TULA-MI IN STRICT BLACK-BOX CASE

**Threat Model.** Given a target sample $x$, a malicious user aims to determine whether $x$ is unlearned data by querying the models before and after unlearning. In a strict black-box scenario, the user inputs $x$ via API and *receives only the output score*, e.g., loss value $\mathcal{L}(x, y)$, quantifying the model's confidence on $x$. Such scenarios are common in real-world applications, such as emotion diagnosis systems that output an emotion label along with a confidence score based on dialogues. Besides, the adversary could easily detect the oc-

---

**Algorithm 2** TULA-MI in the Strict Case

> **Inputs:** Black-box models $\mathcal{M}_{original}$, $\mathcal{M}_{unlearn}$, target sample $(x, y)$, score function $\mathcal{S}$, threshold $\gamma$.
> **if** $|\mathcal{S}(\mathcal{M}_{unlearn}(x), y) - \mathcal{S}(\mathcal{M}_{original}(x), y)| > \gamma$ **then**
>     **Return** $(x, y)$ is an unlearned sample
> **else**
>     **Return** $(x, y)$ is an unseen sample
> **end if**

---

currence of unlearning in practice by continuously querying the model with target samples. If the outputs from two consecutive sets of queries change, the adversary could execute the attack to infer whether any samples have been unlearned. The strict black-box case represents the *minimal knowledge* an adversary can access in practice, thereby indicating the *lower bound* of potential privacy leakage.

**Method.** Typically, the model yields higher confidence for training samples compared to unseen ones. For example, in terms of loss value, $\mathcal{M}_{original}$ exhibits a lower loss for an unlearned sample, while $\mathcal{M}_{unlearn}$ produces a higher one. In contrast, for an unseen sample, both models maintain similarly high loss values. This observation highlights that *unlearned samples undergo significant confidences changes before and after unlearning*, whereas unseen samples generally do not. Therefore, we propose *TULA-MI in the strict case*, leveraging confidence variations to infer the membership information of target samples. The adversary evaluates the confidence change of a target sample before and after unlearning using a threshold $\gamma$, $|\mathcal{S}(\mathcal{M}_{unlearn}(x), y) - \mathcal{S}(\mathcal{M}_{original}(x), y)|$. If the change exceeds $\gamma$, $x$ is likely to be an unlearned sample. The detailed procedure is outlined in Algorithm 2.

#### 4.2.2. TULA-MI IN RELAXED BLACK-BOX CASE

**Threat Model.** Previous MIAs typically assume a relaxed black-box scenario (Chen et al., 2021; Carlini et al., 2022). Specifically, the adversary could obtain the model's logits via queries, additionally possess an identically distributed auxiliary dataset, and know the model architecture. This case reflects the *maximum knowledge* an adversary can obtain in black-box scenarios, thereby indicating the *upper bound* of potential privacy leakage.

**Method.** In this setting, a powerful adversary should be capable of performing *targeted MIA for each sample*. Ideally, the adversary determines the membership of $x$ by carefully fitting two distributions to its output features: the distributions when $x$ is an unlearned sample ($D^x_{unlearn}$) or an unseen sample ($D^x_{unseen}$). To achieve this goal, we decompose the process into three steps: *(1) Feature Space Construction:* For the target sample $x$, we construct the feature vector $[l_{ori}, l_{ul}, l_{ori} - l_{ul}]$ based on its logits from models before and after unlearning, denoted as $l_{ori}$ and $l_{ul}$, respectively. The term $l_{ori} - l_{ul}$ serves as an augmentation to capture logit changes more effectively. *(2) Distribution Esti-*

*mation:* We first sample a training set $\mathcal{D}$ from the auxiliary dataset, train shadow model $\mathcal{M}_{original}$ on $\mathcal{D} \cup x$, and obtain $\mathcal{M}_{unlearn}$ after unlearning $x$. The two models are then queried with $x$ to construct the output feature. By repeating this process multiple times, we estimate the $D^x_{unlearn}$ for $x$. For $D^x_{unseen}$, we sample another training set $\mathcal{D}'$ but exclude $x$, train $\mathcal{M}'_{original}$ on $\mathcal{D}$, and obtain $\mathcal{M}'_{unlearn}$ after randomly unlearning a sample $x'$. Similarly, we can obtain $D^x_{unseen}$ after multiple rounds of sampling and querying. *(3) Attack Model Training:* Features sampled from the two distributions are labeled as 1 and 0, respectively. A classifier is then trained to infer the membership of $x$. The detailed procedure is outlined in Algorithm 4 in Appendix G.

## 4.3. TULA for Data Reconstruction

In this subsection, we introduce TULA-DR, where a *malicious collaborator* aims to reconstruct the unlearned texts with white-box model access.

### 4.3.1. THREAT MODEL

In the white-box scenario, the adversary has access to the model weights both before ($\theta_{original}$) and after ($\theta_{unlearn}$) unlearning and aims to reconstruct the unlearned samples. Additionally, we assume the adversary knows the unlearning algorithm $U$ employed by the model developer. This assumption could be practical. First, the unlearning algorithm itself is not highly confidential, thereby posing a risk of disclosure (e.g., via a complicit developer). Second, model developers may be obligated to disclose details of the unlearning algorithm, performance, and other relevant information to collaborators to uphold transparency and ensure informed updates regarding model versions.

### 4.3.2. THE PROPOSED METHOD

Suppose that an unlearned sample consists of text $x$ and label $y$, where $x$ is a sequence of $L$ tokens ($\langle tk_1, \ldots, tk_L \rangle$) with each $tk_l$ representing a number in the vocabulary $\mathcal{V}$, and $y \in \mathbb{R}^{1 \times C}$ corresponds to one of $C$ categories. Intuitively, the adversary *randomly selects* $L$ tokens from $\mathcal{V}$ and guesses a label $c$ among $C$ categories. The guessed sample is then evaluated by applying the unlearning algorithm $U$ to check whether it enables updating $\theta_{original}$ to $\theta_{unlearn}$. However, this approach is computationally infeasible: (1) The text is represented in a large discrete space, making the search challenging. The adversary faces the task of identifying the target unlearned sample from $C \times L^{|\mathcal{V}|}$ candidates, which is practically intractable. (2) The search process is poorly guided, as the weight differences before and after unlearning are not well incorporated. (3) The lack of constraints for effective search.

Therefore, we propose TULA-DR, a method that efficiently reconstructs unlearned samples by exploiting weight differ-

---

**Algorithm 3** TULA-DR

**Inputs:** original weights $\theta_{original}$, unlearned weights $\theta_{unlearn}$, unlearning algorithm $U$, token embedding dimension $\mathbb{R}^{1 \times d}$, learning rate $\alpha$, regularization factor $\beta$, the maximum number of iterations $N$.

$\hat{x} \sim \mathbb{R}^{L \times d}, \hat{y} \sim \mathbb{R}^{1 \times C}$    candidate random initialization
$\Delta\theta \leftarrow \theta_{original} - \theta_{unlearn}$
**while** not reaching $N$ **do**

$\quad \nabla^U_{\theta_{original}}(\hat{x}, \hat{y}) \leftarrow \frac{\partial U(\theta_{original}, (\hat{x}, \hat{y}))}{\partial \theta_{original}}$

$\quad \mathcal{L}_{rec} \leftarrow 1 - \frac{\nabla^U_{\theta_{\text{original}}}(\hat{x}, \hat{y}) \cdot \Delta\theta}{\left\| \nabla^U_{\theta_{\text{original}}}(\hat{x}, \hat{y}) \right\|_2 \|\Delta\theta\|_2}$

$\quad \mathcal{L}_{reg} \leftarrow \left( \frac{1}{n} \sum_{i=1}^n \|\hat{x}_i\|_2 - \frac{1}{\mathcal{V}} \sum_{j=1}^{\mathcal{V}} \|e_j\|_2 \right)^2$

$\quad \hat{x} \leftarrow \hat{x} + \alpha \cdot \frac{\partial (\mathcal{L}_{rec} + \beta \cdot \mathcal{L}_{reg})}{\partial \hat{x}}$

$\quad \hat{y} \leftarrow \hat{y} + \alpha \cdot \frac{\partial (\mathcal{L}_{rec} + \beta \cdot \mathcal{L}_{reg})}{\partial \hat{y}}$

$\quad \text{Clip}(\hat{x}; x_{low}, x_{up})$
**end while**
**Return** Decoding$(\hat{x}, \hat{y})$

---

ences before and after unlearning:

**Candidate Initialization:** To address the challenge (1), we reformulate the discrete optimization problem as a continuous one. Specifically, assuming the token embedding dimension $\mathbb{R}^{1 \times d}$ and a token embedding layer $W_{wordEbd} : \mathbb{R}^{|\mathcal{V}| \times d}$, a tokenized text $x$ can be represented with an embedding vector in $\mathbb{R}^{L \times d}$. Such transformation significantly reduces the search space as $d \ll |\mathcal{V}|$. Additionally, gradient-based optimizers (e.g., Adam (Kingma & Ba, 2014)) can be utilized to efficiently search the target text.

**Reconstruction Objective:** To address the challenge (2), we design a loss function that guides the optimization by *mirroring the unlearning process and approximating the weight differences*. Inspired by (Hu et al., 2024) the gradient of the unlearned samples over the original model aligns with the difference in model weights before and after unlearning ($\Delta\theta$). Consequently, we compute the gradient of the candidate ($\hat{x}, \hat{y}$) on $\mathcal{M}_{original}$ executing the unlearning algorithm $U$, and then use *cosine similarity* to compute the angle between the gradient and weight difference, which is utilized as the loss to update candidate:

$$\mathcal{L}_{rec} = 1 - \frac{\nabla^U_{\theta_{\text{original}}}(\hat{x}, \hat{y}) \cdot \Delta\theta}{\left\| \nabla^U_{\theta_{\text{original}}}(\hat{x}, \hat{y}) \right\|_2 \|\Delta\theta\|_2}. \quad (1)$$

**Embedding Constraints:** To address the challenge (3), we introduce *sentence regularization* and *boundary clipping* mechanisms to constrain the distribution of reconstructed embeddings and accelerate convergence. The word embedding layer $W_{wordEbd}$ pre-stores all possible candidate tokens in a vocabulary. Based on this intuition, we first introduce a *sentence regularization* term that enforces the norm of reconstructed sentence embeddings to approximate the average norm of the vocabulary (Balunovic et al., 2022),

preventing the optimization process from local optima:

$$\mathcal{L}_{reg} = \left( \frac{1}{n} \sum_{i=1}^{n} \|\hat{x}_i\|_2 - \frac{1}{\mathcal{V}} \sum_{j=1}^{\mathcal{V}} \|e_j\|_2 \right)^2. \quad (2)$$

Additionally, we propose a *boundary clipping* mechanism to ensure that each embedding element remains within the maximum and minimum values of all potential tokens at the corresponding element position, thereby calibrating the optimization direction and accelerating convergence:

$$\text{Clip}(\hat{x}; x_{low}, x_{up}) = \min \left( \max \left( \hat{x}, x_{low} \right), x_{up} \right), \quad (3)$$

where $x_{low}$ and $x_{up}$ are calculated in $W_{wordEbd}$. The detailed procedure is outlined in Algorithm 3. Besides, the Decoding$(\cdot)$ function refers to transforming the reconstructed embeddings into text space. Specifically, we use the model's embedding layer as a "vocabulary" and apply cosine similarity to match the indices of the embeddings and then convert them into readable texts via the tokenizer.

# 5. Experiments

## 5.1. Experimental Setups

**Datasets:** In this paper, we investigate unlearning mechanisms on fine-tuned LMs, since the fine-tuning datasets are more configurable and practical for evaluations (Maini et al., 2024). We manually construct two *synthetic datasets*, *SynthPAI-age* and *SynthPAI-inc*, for the evaluations. They are two binary text classification datasets for attribute inference tasks, where texts consist of comments generated by GPT-4-based agents during interactions, labels are commenters' attributes, such as *age* (*young* or *old*) and *income* (*low* or *high*). A detailed introduction on datasets is presented in Appendix A. Besides, we focus on the text classification tasks, because: (1) they enable us to design rigorous audit samples for rigorous unlearning auditing in the worst case. Specifically, the utilized mislabeled samples have been found to be most vulnerable to privacy leakage on image classification tasks (Aerni et al., 2024), and rigorous samples for text generation tasks are currently under-explored. (2) They ensure a fair evaluation on MIA-based auditing and attack methods (Pawelczyk et al., 2023), as MIA remains poorly defined and lacks robust metrics for text generation tasks (Duan et al., 2024).

**Why synthetic datasets:** Previous studies indicate that LMs might have "already seen" the fine-tuned samples in public datasets during pre-training, potentially undermining the reliability of unlearning processes (Maini et al., 2024). In addition, the real-world data can not meet our requirement, since: (1) pre-trained models are trained on vast amounts of real-world data, which is typically closed-sourced. (2) Recent studies indicate that we can not yet robustly detect

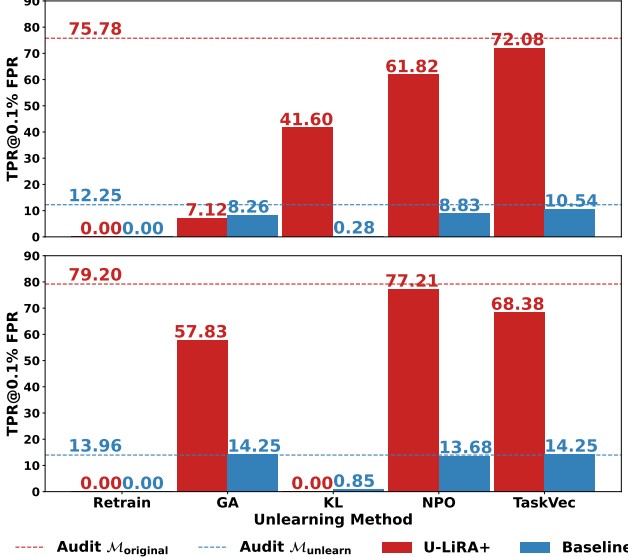

*Figure 3.* Auditing unlearning methods on *SynthPAI-inc* dataset. The lines and bars represent the auditing results on the original models and corresponding unlearned models, respectively. OPT-1.3b model (Top), Pythia-1.4b model (Bottom).

whether a specific sample was used among massive pre-training data (Zhang et al., 2024a). Therefore, we utilize synthetic datasets to rigorously ensure the fine-tuning on previously "unseen" data. Specifically, the synthetic samples are entirely based on *fictional scenarios*, and generated by GPT-4-based agents *released after the models we utilized*. Through these efforts, we hope to ensure that the setups are rigorous and thus avoid being potentially misleading.

**Models:** We employ Pythia-1.4b (Biderman et al., 2023) and OPT-1.3b (Zhang et al., 2022) LMs for evaluations. To ensure effective learning, we apply full-parameter fine-tuning and meticulously configure the training process to prevent overfitting. Detailed training configurations and performance results are provided in Appendix B.

**Unlearning Methods:** We conduct a comprehensive investigation on both *exact* and *inexact* unlearning mechanisms. For *exact unlearning*, we examine the retraining method (*Retrain*). For *inexact unlearning*, we evaluate four widely adopted methods: gradient ascent (*GA*), maximizing Kullback-Leibler divergence (*KL*), negative preference optimization (*NPO*), and task vector (*TaskVec*). To ensure erasure is as thorough as possible, we apply full-parameter fine-tuning and carefully configure each method. Detailed descriptions and configurations are provided in Appendix C.

## 5.2. Evaluations on Textual Unlearning Auditing

In this subsection, we evaluate our proposed auditing method, U-LiRA+, across five unlearning methods on

*Table 1.* TULA-MI in strict black-box cases. The metric is averaged $NTS@1FS$, and the presented results are evaluated on the Pythia-1.4b model and SynthPAI-age dataset.

| N | ScoreFunc | Retrain | GA | KL | NPO | TaskVec |
|---|-----------|---------|-----|-----|-----|---------|
| 32 | Confidence | 4.7 | 6.45 | 4.65 | 4.6 | 6.35 |
| | CrossEntropy | 11.4 | 4.75 | 7.6 | 4.3 | 4.75 |
| | Hinge | 4.6 | 7.4 | 4.35 | 9.5 | 7.35 |
| 64 | Confidence | 4.55 | 14.25 | 10.2 | 10.55 | 13.95 |
| | CrossEntropy | 27.55 | 4.7 | 10.1 | 5.85 | 4.7 |
| | Hinge | 4.45 | 18.2 | 8.8 | 21.75 | 18.05 |

*SynthPAI-inc* dataset, as shown in Figure 3. We employ the SOTA auditing method U-LiRA as a baseline (Hayes et al., 2024). Detailed implementation of U-LiRA+ and results on *SynthPAI-age* dataset are provided in Appendix D. We default the unlearning ratio to $1\%$. We employ the True-Positive Rate at a low False-Positive Rate ($TPR@0.1\%FPR$) as the audit metric, reflecting how confidently an adversary compromises individual privacy. For $\mathcal{M}_{unlearn}$, a lower value indicates that the adversary has less confidence in distinguishing audit samples (unlearned and unseen samples), implying more successful unlearning.

Through rigorous auditing of existing unlearning methods, our empirical results reveal that the current auditing method significantly overestimates the effectiveness of unlearning mechanisms. For instance, in the audit results of *KL* method on the OPT-1.3b model in Figure 3, the baseline suggests that *KL* substantially reduces TPR on the unlearned model (from $12.25\%$ to $0.28\%$). However, our U-LiRA+ method shows that TPR remains at $41.60\%$ on $\mathcal{M}_{unlearn}$, indicating that the adversary can still effectively distinguish audit samples after unlearning. Moreover, while *Retrain* could achieve a thorough erasure, inexact unlearning methods typically present a false sense of unlearning. Across various settings, inexact unlearning methods fail to effectively erase audited samples compared to *Retrain*, as evidenced by the consistently high TPR observed with U-LiRA+.

### 5.3. Exploring Black-Box Privacy Risks against Textual Unlearning Systems

In this subsection, we evaluate the proposed TULA-MI against textual unlearning systems when the adversary has black-box access to the models before and after unlearning.

#### 5.3.1. TULA-MI IN STRICT BLACK-BOX CASE

We begin with the strict black-box scenarios where the adversary can only access *output scores*. To ensure a comprehensive evaluation, we examine three commonly used score functions: confidence ($f(x)_y$), cross-entropy loss ($-log((x)_y)$), and hinge loss ($log(p/(1-p)), p = f(x)_y$). Additionally, we consider two unlearning ratios, $0.5\%$ and $1\%$ (the number of audited samples $N = 32, 64$ for

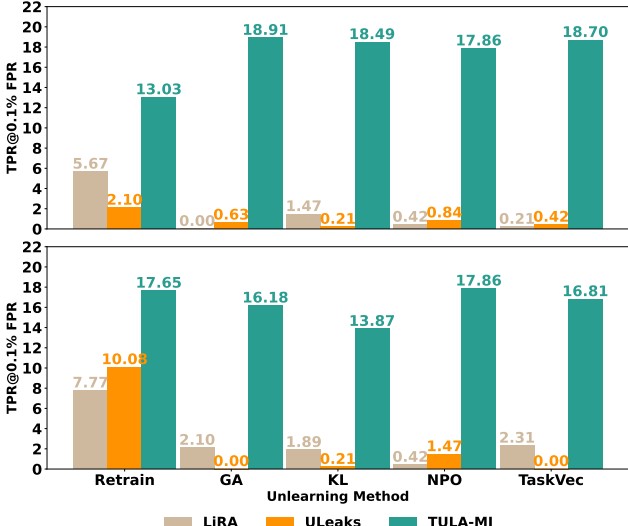

*Figure 4.* MIAs in relaxed black-box case on *SynthPAI-age* dataset. OPT-1.3b model (Top), Pythia-1.4b model (Bottom).

*SynthPAI-age* and $24, 48$ for *SynthPAI-inc*). Besides, we incorporate the metric $NTS@1FS$ (Number of True Samples @ 1 False Sample), which quantifies the number of correctly inferred samples with only one mistake.

We present the results on Pythia-1.4b and *SynthPAI-age* dataset in Table 1, and results in other settings are provided in Appendix E. The results reveal that even with only access to output scores, the adversary can effectively infer membership information of the unlearned samples, with several samples correctly identified while just one incorrect. Moreover, *Retrain* is more vulnerable than inexact unlearning methods in such cases. As shown in Table 1, with a single incorrect guess, the adversary can infer more than 27 correct samples against the retrained model.

#### 5.3.2. TULA-MI IN RELAXED BLACK-BOX CASE

We then move to the relaxed black-box cases, where the adversary can query the model to obtain logit vectors, is aware of the model architecture, and has access to an identically distributed auxiliary dataset. Under these conditions, we evaluate two baseline methods: LiRA (Carlini et al., 2022) and ULeaks (Chen et al., 2021). Detailed descriptions and implementations of these methods are provided in Appendix G.

As shown in Figure 4, we present the MIA results on *SynthPAI-age* dataset, and additional results on *SynthPAI-inc* dataset are included in Appendix F. With a wrong guess below $0.1\%$, TULA-MI accurately infers membership for over $18\%$ of the samples, highlighting significant privacy risks. Compared to ULeaks, our method achieves superior performance by employing a more accurate per-sample at-

*Table 2.* Text reconstruction performance of TULA-DR.

| Model | Method | SynthPAI-age | | | SynthPAI-inc | | |
|---|---|---|---|---|---|---|---|
| | | R-1 | R-2 | R-L | R-1 | R-2 | R-L |
| Pythia-1.4b | GA | 81.04 | 66.75 | 81.35 | 81.92 | 58.16 | 75.15 |
| | KL | 78.8 | 43.29 | 71.18 | 65.8 | 43.58 | 65.47 |
| | NPO | 69.61 | 32.31 | 64.42 | 74.27 | 44.43 | 69.54 |
| | TaskVec | 81.94 | 65.59 | 65.6 | 72.63 | 41.24 | 67.8 |
| OPT-1.3b | GA | 43.57 | 17.5 | 40.36 | 38.05 | 10.28 | 35.13 |
| | KL | 15.06 | 1.82 | 12.83 | 17.35 | 4.25 | 16.46 |
| | NPO | 32.1 | 10.69 | 30.81 | 31.28 | 7.3 | 26.61 |
| | TaskVec | 41.27 | 16.91 | 38.18 | 46.95 | 22 | 40.29 |

tack strategy. While LiRA also utilizes a per-sample attack and fits normal distributions for unlearned and unseen outputs, its effectiveness is hindered by redundancy and overlap among textual samples, making it challenging to fit independent distributions for reliable discrimination. In contrast, our method explicitly learns complex decision boundaries for target samples in unlearned and unseen cases, enabling more robust MIA.

### 5.4. Exploring White-Box Privacy Risks against Textual Unlearning Systems

In this subsection, we examine the effectiveness of TULA-DR with white-box access to model weights before and after unlearning. To ensure a fair evaluation, we employ three metrics to assess reconstruction performance: ROUGE-1 (R-1), ROUGE-2 (R-2), and ROUGE-L (R-L) (Lin, 2004), which respectively measure the ratios of reconstructed unigrams, bigrams, and the longest-matching subsequence. Additionally, reconstructed examples are provided in Appendix J. We assume a default scenario where $\mathcal{M}_{original}$ is trained on 1000 samples, with one sample subsequently unlearned. Furthermore, our method can be extended to batch unlearning scenarios, and the results are presented in Appendix H. For TULA-DR, the learning rate $\alpha$ is set to 0.1 for the Pythia-1.4b model and 0.45 for the OPT-1.3b model, and the regularization factor $\beta$ is fixed at 0.1. Besides, an ablation study is further provided in Appendix I.

As shown in Table 2, discrepancies in unlearning weights can result in substantial textual privacy leakage. Specifically, for the Pythia-1.4b model, over 80% of the original words were accurately reconstructed, as the R-1 values show. Besides, the reconstructed sentences preserved both semantic structure and meaning with high R-2 and R-L values, respectively. Besides, for the OPT-1.3b model, although the reconstructed sentences exhibited disordered semantic structure, an average of 40% of the original words were still recovered, exposing much information. Specifically, the adversary could still utilize the reconstructed words to infer the useful information of the original texts.

## 6. Conclusion

In this paper, we investigated textual unlearning methods on LMs, highlighting their vulnerabilities and privacy risks. We introduced a rigorous unlearning auditing method, U-LiRA+, providing an accurate assessment of erasure effectiveness. Additionally, to explore the privacy risks of textual unlearning mechanisms, we proposed TULA and its variants in black- and white-box scenarios: TULA-MI and TULA-DR. Through extensive evaluations, our findings revealed that textual unlearning actually gives a false sense of unlearning. Current methods fail to completely erase unlearned texts, which remain highly detectable after unlearning. Moreover, deploying textual unlearning mechanisms could instead increase privacy risks. Specifically, by examining models before and after unlearning, membership information of unlearned texts can be confidently inferred using TULA-MI. In the white-box scenario, they can even be accurately reconstructed using TULA-DR. Our findings indicated that existing textual unlearning methods are less effective and reliable than expected, highlighting the need for more robust and secure unlearning mechanisms on LMs.

### Acknowledgements

This work is supported by National Natural Science Foundation of China (Grants No. U24B20182, 62122066), National Key R&D Program of China (Grant No. 2021ZD0112803), and Key R&D Program of Zhejiang (Grant No. 2022C01018).

### Impact Statement

This paper presents work whose goal is to advance the field of Machine Learning. We reveal the vulnerabilities and privacy risks of existing machine unlearning methods on language models and encourage the development of more robust and secure unlearning mechanisms. There are many potential societal consequences of our work, none of which we feel must be specifically highlighted here.

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

# A. Detailed Description and Discussion On Datasets

To ensure rigorous evaluation, we construct two synthetic datasets derived from the *SynthPAI* dataset (Yukhymenko et al., 2024), enabling fine-tuning on previously "unseen" data. SynthPAI is a synthetic dataset primarily designed for evaluating attribute inference attacks. It leverages multiple GPT-4-based agents simulating commenters with different attributes (e.g., age, gender, income) communicating with each other, aiming to test whether an adversary can infer these attributes based on the comments. From SynthPAI, we create two binary text classification datasets: *SynthPAI-age* and *SynthPAI-inc*. In *SynthPAI-age*, the labels correspond to age groups *(young or old)*, with *young* defined as age $\leq 20$ and *old* as age $> 50$, resulting in a total of 3200 samples. For SynthPAI-inc, we adopt the original *SynthPAI* partition for income levels (*low* and *high*), yielding a total of 2400 samples.

*Table 3.* Memory reconstruction results on the utilized models with the proposed datasets. ROUGE-1 (R-1), ROUGE-2 (R-2), and ROUGE-L (R-L). And lower ROUGE values indicate the generated texts are less similar to the ground-truth texts.

| Model | SynthPAI-age | | | | SynthPAI-inc | | | |
|---|---|---|---|---|---|---|---|---|
| | R-1 | R-2 | R-L | Loss | R-1 | R-2 | R-L | Loss |
| Pythia-1.4b | 0.047 | 0.003 | 0.046 | 5.68 | 0.067 | 0 | 0.063 | 5.92 |
| OPT-1.3b | 0.055 | 0.001 | 0.051 | 5.54 | 0.046 | 0 | 0.045 | 5.69 |

Since *SynthPAI* consists of synthetic samples in fictional scenarios, and GPT-4 was released after the LMs used in our experiments, our proposed datasets maintain considerable rigor. To further validate this, we conduct memory extraction attacks (Carlini et al., 2021) on the models using our proposed datasets. Specifically, we select 100 samples from the datasets and use 50% of a sample as a prompt to evaluate the models' ability to generate the remaining 50%. As shown in Table 3, the ROUGE values are very low (generated texts are nearly random guesses), indicating that the models have not been trained on the synthetic samples.

# B. Training Configurations

We utilize the *AdamW* optimizer (Loshchilov, 2017) to full-parameter fine-tune the model to obtain $\mathcal{M}_{original}$, with the learning rate set to $1e^{-5}$, the batch size to 64, and the number of training rounds to 2. We carefully prevent the models from overfitting and further provide *train accuracy* and *test accuracy* in Table 4.

*Table 4.* Training accuracy and test accuracy of $\mathcal{M}_{original}$.

| Model | SynthPAI-age | | SynthPAI-inc | |
|---|---|---|---|---|
| | Train ACC | Test ACC | Train ACC | Test ACC |
| Pythia-1.4b | 95.21% | 85.31% | 93.68% | 89.06% |
| OPT-1.3b | 95.11% | 86.85% | 93.53% | 89.27% |

# C. Introduction to Unlearning Methods and Configurations

In this part, we present the details of the textual unlearning methods we utilized along with their configurations. All the unlearning methods are implemented via full-parameter fine-tuning on the target LMs to ensure the effectiveness of erasing.

• **Retrain:** remove the unlearned samples ($\mathcal{D}_{unlearn}$) from the training set ($\mathcal{D}$) and retrain the original model ($\mathcal{M}_{original}$) on the remaining data ($\mathcal{D}_{remian}$) from scratch, which provides the cleanest erase and is considered as the "gold standard" (Bourtoule et al., 2021). The retrain setting is kept the same as the original fine-tuning.

• **Gradient Ascent (GA):** minimizes the likelihood of predictions on the unlearned samples ($\mathcal{D}_{unlearn}$) by performing gradient ascent on the cross-entropy loss (the opposite of gradient descent) (Jang et al., 2022; Shi et al., 2024). The unlearning batch size is set to 32 for *SynthPAI-age* and *SynthPAI-inc* datasets, respectively. The unlearning rate is set to $1e^{-4}$ and the unlearning epoch is 4.

• **Maximizing Kullback-Leibler divergence (KL):** maximizes the KL divergence of the outputs between $\mathcal{M}_{original}$ and

unlearned model ($\mathcal{M}_{unlearn}$) on $\mathcal{D}_{unlearn}$ (Chen & Yang, 2023). The unlearning batch size is set to 32 for *SynthPAI-age* and *SynthPAI-inc* datasets, respectively. The unlearning rate is set to $1e^{-4}$ and the unlearning epoch is 1.

• **Negative Preference Optimization (NPO):** treats $\mathcal{D}_{unlearn}$ as negative preference data and adjust the offline DPO objective to fine-tune $\mathcal{M}_{original}$ such that it assigns a low likelihood to $\mathcal{D}_{unlearn}$ (Zhang et al., 2024b).

$$\mathcal{L}_{\mathrm{NPO}}(\mathcal{M}_{unlearn}) = -\frac{2}{\beta}\mathbb{E}_{x \sim \mathcal{D}_{\mathrm{unlearn}}}\left[\log \sigma \left(-\beta \log \frac{\mathcal{M}_{unlearn}}{\mathcal{M}_{original}}\right)\right],$$

where $\sigma$ is the sigmoid function, and $\beta$ is a hyperparameter that controls the allowed divergence of $\mathcal{M}_{unlearn}$ from its initialization $\mathcal{M}_{original}$. Following (Shi et al., 2024), we set $\beta$ to 0.1. The unlearning batch size is set to 32 for *SynthPAI-age* and *SynthPAI-inc* datasets, respectively. The unlearning rate is set to $1e^{-4}$ and the unlearning epoch is 7.

• **Task Vectors (TaskVec):** performs unlearning in two stages (Ilharco et al., 2022). $\mathcal{M}_{original}$ is first overfitted on $\mathcal{D}_{unlearn}$ to create an intermediary model ($\mathcal{M}_{reinforce}$). The unlearned model ($\mathcal{M}_{unlearn}$) is then obtained directly by subtracting the weight differences: $\mathcal{M}_{unlearn} = \mathcal{M}_{original} - (\mathcal{M}_{reinforce} - \mathcal{M}_{original})$. The unlearning batch size is set to 32 for *SynthPAI-age* and *SynthPAI-inc* datasets, respectively. The unlearning rate is set to $1e^{-4}$ and unlearning epoch is 4.

## D. Implementation on U-LiRA+ and Additional Results

We default the unlearning ratio to $1\%$, indicating that $N = 64$ or $48$ audited samples are required for *SynthPAI-age* and *SynthPAI-inc* datasets (half unlearn or unseen samples), respectively. Besides, considering that the datasets have an equal number of 0 or 1 labeled samples, we default all the audited samples are selected from 1-labeled samples. To implement our proposed auditing method, we train 100 shadow models for each audited sample, where $T = 85$ of them are for estimating the distributions of model outputs when the sample is either unlearned or unseen, and 15 of them are acting as tested models $\mathcal{M}^*$ for computing $TPR@LowFPR$ evaluating the audited unlearning methods. Ideally, we need to fine-tune at least $N \times T$ shadow models to complete one time of auditing, which is too much computational and storage overhead for LMs. Therefore, we adopt the optimization scheme mentioned in the original LiRA-MIA (Carlini et al., 2022), requiring only $T$ shadow models, where each model is randomly trained (and then unlearned) on half of the audit samples. Such a scheme enables $T/2$ shadow models for each audit sample to approximate the unlearned and unseen distributions, which significantly reduces the computational overhead while keeping the auditing accuracy as much as possible.

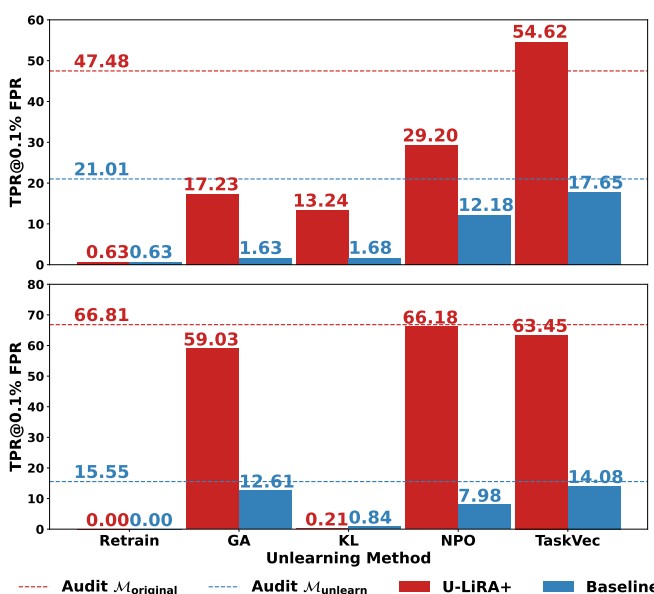

*Figure 5.* Auditing unlearning methods on *SynthPAI-age* dataset. The lines and bars represent the auditing results on the original models and corresponding unlearned models, respectively. OPT-1.3b model (Top), Pythia-1.4b model (Bottom).

# E. Additional Results of TULA-MI in Strict Cases

*Table 5.* TULA-MI in strict black-box cases. The presented results are evaluated on the OPT-1.3b model and SynthPAI-age dataset.

| N | LossFunc | Retrain | GA | KL | NPO | TaskVec |
|---|---|---|---|---|---|---|
| 32 | Confidence | 6.55 | 3.45 | 5.1 | 2.7 | 8.25 |
| | CrossEntropy | 10.35 | 4.25 | 5.8 | 4.4 | 5.9 |
| | Hinge | 5.65 | 1.55 | 4.6 | 1.8 | 8.05 |
| 64 | Confidence | 6.25 | 5.6 | 7.7 | 5.6 | 16.55 |
| | CrossEntropy | 25.1 | 4.15 | 7.1 | 6.35 | 6.95 |
| | Hinge | 6.9 | 0.8 | 6.55 | 1.75 | 17.85 |

*Table 6.* TULA-MI in strict black-box cases. The presented results are evaluated on the OPT-1.3b model and SynthPAI-inc dataset.

| N | LossFunc | Retrain | GA | KL | NPO | TaskVec |
|---|---|---|---|---|---|---|
| 24 | Confidence | 3.75 | 2.35 | 3.5 | 1.8 | 4.15 |
| | CrossEntropy | 8.2 | 3.1 | 5.3 | 2.8 | 4.55 |
| | Hinge | 3.8 | 1.6 | 3.55 | 2.35 | 4.85 |
| 48 | Confidence | 9.1 | 3.5 | 5.65 | 2.75 | 10.55 |
| | CrossEntropy | 16.25 | 4.95 | 8.4 | 4.5 | 8.05 |
| | Hinge | 7.75 | 1 | 7.4 | 2.5 | 11.5 |

*Table 7.* TULA-MI in strict black-box cases. The presented results are evaluated on the Pythia-1.4b model and SynthPAI-inc dataset.

| N | LossFunc | Retrain | GA | KL | NPO | TaskVec |
|---|---|---|---|---|---|---|
| 24 | Confidence | 4.55 | 5.65 | 4.85 | 4.15 | 5.65 |
| | CrossEntropy | 7.6 | 4.55 | 4.9 | 6.75 | 4.55 |
| | Hinge | 4.4 | 5.3 | 3.25 | 8.8 | 5.2 |
| 48 | Confidence | 9.6 | 9.8 | 10.55 | 9.55 | 9.7 |
| | CrossEntropy | 14.7 | 9.5 | 9.3 | 14.05 | 9.5 |
| | Hinge | 9.3 | 9.85 | 7.85 | 15.85 | 9.8 |

# F. Additional Results of TULA-MI in Relaxed Cases

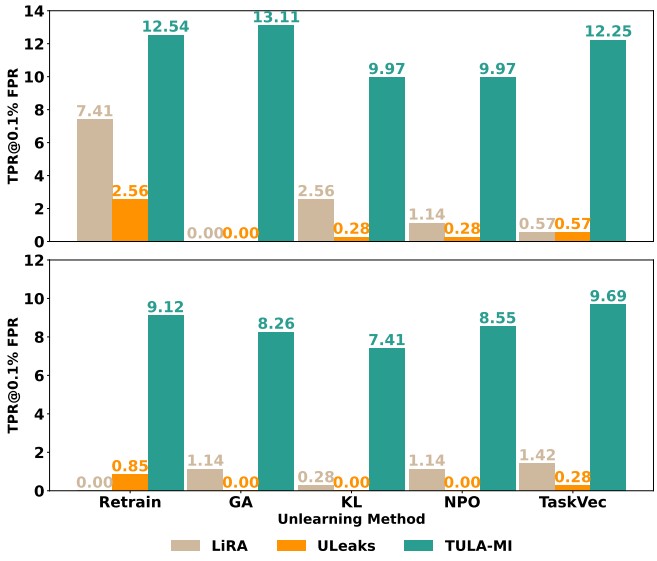

*Figure 6.* MIAs in relaxed black-box case on *SynthPAI-inc* dataset. OPT-1.3b model (Top), Pythia-1.4b model (Bottom).

## G. Detailed Implementation of TULA-MI in Relaxed Cases

We train 100 shadow models to implement the baseline *LiRA* (Carlini et al., 2022), as referred in Appendix D. Besides, to implement *ULeaks* (Chen et al., 2021), we follow its original settings and utilize a *random forest* classifier as the attack model, with $n\_estimators = 500$ and $min\_samples\_leaf = 30$. We train the attack model with the output features on shadow models, where the logits of unlearned samples on $\mathcal{M}_{unleanr}$ and $\mathcal{M}_{original}$ are combined as positive features, and these of unseen samples are utilized as negative features. To implement our proposed *TULA-MI*, we utilize *LightGBM* classifier (Ke et al., 2017) as the attack model, with $num\_leaves = 2$, $learning\_rate = 0.05$ and $feature\_fraction = 0.9$.

---

**Algorithm 4** TULA-MI in the Relaxed Case

---

**Inputs:** original black-box model $\mathcal{M}^*_{original}$, unlearned black-box model $\mathcal{M}^*_{unlearn}$, target sample $(x, y)$,random sample $(x', y')$, auxiliary dataset $\mathcal{D}_{aux}$, learning algorithm $A$, unlearning algorithm $U$, number of shadow models $T$, logit function $\phi$.
$D^x_{unlearn} \leftarrow \{\}, D^x_{original} \leftarrow \{\}$
**while** $t \leq T$ **do**
  $\mathcal{D} \sim \mathcal{D}_{aux}$   sample a shadow training dataset
  $\mathcal{M}_{original} \leftarrow A(\mathcal{D} \cup (x, y))$
  $\mathcal{M}_{unlearn} \leftarrow U(\mathcal{M}_{original}, (x, y)))$
  $l^x_{ori} \leftarrow \phi((x, y); \mathcal{M}_{original}), l^x_{ul} \leftarrow \phi((x, y); \mathcal{M}_{unlearn})$
  $D^x_{unlearn}[t] \leftarrow ([l^x_{ori}, l^x_{ul}, l^x_{ori} - l^x_{ul}], 1)$   labeled with 1
  $\mathcal{D}' \sim \mathcal{D}_{aux}$   sample another shadow dataset excluding $x$
  $\mathcal{M}'_{original} \leftarrow A(\mathcal{D}')$
  $\mathcal{M}'_{unlearn} \leftarrow U(\mathcal{M}'_{original}, (x', y')))$
  $l^x_{ori'} \leftarrow \phi((x, y); \mathcal{M}'_{ori}), l^x_{ul'} \leftarrow \phi((x, y); \mathcal{M}'_{unlearn})$
  $D^x_{unseen}[t] \leftarrow ([l^x_{ori'}, l^x_{ul'}, l^x_{ori'} - l^x_{ul'}], 0)$   labeled with 0
**end while**
$\mathcal{M}_{Adv} \leftarrow$   train attack model on $D^x_{unlearn} \cup D^x_{unseen}$
$l^x_{ori*} \leftarrow \phi((x, y); \mathcal{M}^*_{original}), l^x_{ul*} \leftarrow \phi((x, y); \mathcal{M}^*_{unlearn})$
**Return** $\mathcal{M}_{Adv}([l^x_{ori*}, l^x_{ul*}, l^x_{ori*} - l^x_{ul*}])$

---

## H. Evaluations on TULA-DR against Batch Unlearning

In this section, we evaluate the reconstruction performance of the proposed TULA-DR compared to batch unlearning, as presented in Tables 8, 9, 10, 11. TULA-DR retains the ability to reconstruct certain sensitive information even from batched unlearned texts, particularly when using the Pythia-1.4b model. Furthermore, the reconstruction accuracy decreases with an increase in batch size ($B$). This decline is attributed to the expansion of the embedding space, which complicates the search process. Notably, as batch size increases, the R-2 score of the reconstructed text declines more significantly than the R-1 and R-L scores. This indicates that batch unlearning effectively reduces an adversary's ability to reconstruct the semantic structure of a sentence, although keywords and word order can still be partially reconstructed.

*Table 8.* TULA-DR against batch unlearning (*GA*).

| Model | B | SynthPAI-age | | | SynthPAI-inc | | |
|---|---|---|---|---|---|---|---|
| | | R-1 | R-2 | R-L | R-1 | R-2 | R-L |
| Pythia-1.4b | 2 | 32.54 | 16.66 | 28.83 | 39.1 | 23.28 | 37.91 |
| | 4 | 28.43 | 9.32 | 26.31 | 28.32 | 7.73 | 27.62 |
| | 6 | 32.27 | 16.18 | 30.52 | 28.35 | 8.7 | 26.48 |
| | 8 | 18.63 | 2.95 | 18.09 | 24.38 | 6.61 | 23.27 |
| OPT-1.3b | 2 | 12.41 | 0 | 8.7 | 19.9 | 4.76 | 17.01 |
| | 4 | 15.8 | 0 | 12.61 | 20.07 | 3.42 | 18.18 |
| | 6 | 17.92 | 1.62 | 16.1 | 16.12 | 1.72 | 15.3 |
| | 8 | 14.94 | 2.04 | 14.14 | 13.13 | 1.04 | 12.63 |

*Table 9.* TULA-DR against batch unlearning (*KL*).

| Model | B | SynthPAI-age | | | SynthPAI-inc | | |
|---|---|---|---|---|---|---|---|
| | | R-1 | R-2 | R-L | R-1 | R-2 | R-L |
| Pythia-1.4b | 2 | 57.93 | 28.57 | 48.31 | 40.23 | 21.69 | 40.23 |
| | 4 | 33.92 | 23.82 | 34.06 | 30.8 | 8.61 | 28.03 |
| | 6 | 25.58 | 9.04 | 26.07 | 24.54 | 4.07 | 23.92 |
| | 8 | 18.85 | 5.506 | 16.96 | 28.03 | 5.456 | 20.55 |
| OPT-1.3b | 2 | 12.03 | 1.85 | 10.37 | 15.18 | 2.38 | 15.18 |
| | 4 | 14.31 | 0 | 14.48 | 15 | 2.89 | 14.02 |
| | 6 | 11.34 | 1.38 | 11.4 | 13.55 | 1.66 | 12.98 |
| | 8 | 14.37 | 1.44 | 13.71 | 15.75 | 1.5 | 15.19 |

*Table 10.* TULA-DR against batch unlearning (*NPO*).

| Model | B | SynthPAI-age | | | SynthPAI-inc | | |
|---|---|---|---|---|---|---|---|
| | | R-1 | R-2 | R-L | R-1 | R-2 | R-L |
| Pythia-1.4b | 2 | 47.91 | 24 | 43.15 | 45.86 | 7.93 | 45.86 |
| | 4 | 32.13 | 9.37 | 29.32 | 20.02 | 5.83 | 19 |
| | 6 | 23.02 | 2.03 | 20.11 | 30.85 | 13.82 | 31.53 |
| | 8 | 29.13 | 7.29 | 28.5 | 24.96 | 5.76 | 21.09 |
| OPT-1.3b | 2 | 16.2 | 4.16 | 16.01 | 23.06 | 4.16 | 23.06 |
| | 4 | 18.65 | 0 | 14.73 | 15.08 | 2.02 | 14.16 |
| | 6 | 13.19 | 1.23 | 11.94 | 11.78 | 3.51 | 12.12 |
| | 8 | 13.57 | 0.52 | 12.14 | 13 | 3.24 | 13 |

*Table 11.* TULA-DR against batch unlearning (*TaskVec*).

| Model | B | SynthPAI-age | | | SynthPAI-inc | | |
|---|---|---|---|---|---|---|---|
| | | R-1 | R-2 | R-L | R-1 | R-2 | R-L |
| Pythia-1.4b | 2 | 24.68 | 7.4 | 23.01 | 44.82 | 11.31 | 41.54 |
| | 4 | 27.97 | 9.72 | 26.88 | 26.99 | 8.05 | 26.51 |
| | 6 | 23.91 | 9.92 | 21.34 | 27.05 | 8.73 | 25.79 |
| | 8 | 21.97 | 7.5 | 20.25 | 24.18 | 10.78 | 23.02 |
| OPT-1.3b | 2 | 23.28 | 6.25 | 23.28 | 30.45 | 9.25 | 28.14 |
| | 4 | 19.62 | 1.04 | 18.47 | 16.93 | 4.79 | 16.93 |
| | 6 | 14.04 | 2.42 | 13.22 | 11.24 | 1.58 | 11.29 |
| | 8 | 13.11 | 0.98 | 10.76 | 9.87 | 1.11 | 9.17 |

## I. Ablation Study on TULA-DR

In this section, we perform an ablation study on TULA-DR to analyze its design components. Four settings are compared, representing different configurations to reconstruct the unlearned texts: ($\mathcal{L}_{rec}$) using only cosine distance, ($\mathcal{L}_{rec}(l_2$-based)) using only Euclidean distance, ($\mathcal{L}_{rec} + \mathcal{L}_{reg}$) combining cosine distance with $\mathcal{L}_{reg}$ *sentence regularization*, and ($\mathcal{L}_{rec} + Clip$) combining cosine distance with the *boundary clipping* mechanism.

First, cosine distance demonstrates superior reconstruction performance compared to Euclidean distance. This advantage arises because cosine distance captures directional similarity in high-dimensional model weight vectors more effectively, facilitating better guidance for reconstructing unlearned texts, whereas Euclidean distance is harder to optimize in such cases. Furthermore, incorporating $\mathcal{L}_{reg}$ *sentence regularization* significantly enhances reconstruction accuracy by providing additional statistical information (mean embedding values), which helps the reconstructed embeddings align more closely with the correct distribution. Lastly, adding the *boundary clipping* mechanism also improves reconstruction effectiveness. By calibrating the value domain of reconstructed embeddings during optimization, this mechanism accelerates convergence and prevents falling into local optima.

*Table 12.* Ablation study on TULA-DR against *GA*.

| Model | Setting | SynthPAI-age | | | SynthPAI-inc | | |
|---|---|---|---|---|---|---|---|
| | | R-1 | R-2 | R-L | R-1 | R-2 | R-L |
| Pythia-1.4b | $\mathcal{L}_{rec}$ | 66.66 | 25 | 59.25 | 29.76 | 4.76 | 29.76 |
| | $\mathcal{L}_{rec}(l_2\text{-based})$ | 0 | 0 | 0 | 0 | 0 | 0 |
| | $\mathcal{L}_{rec} + \mathcal{L}_{reg}$ | 80.15 | 36.66 | 70.63 | 66.31 | 34.12 | 62.97 |
| | $\mathcal{L}_{rec} + Clip$ | 92.59 | 62.5 | 84.12 | 80.55 | 60.11 | 76.38 |
| OPT-1.3b | $\mathcal{L}_{rec}$ | 6.061 | 0 | 6.061 | 16.36 | 0 | 16.36 |
| | $\mathcal{L}_{rec}(l_2\text{-based})$ | 11.57 | 0 | 11.57 | 5.55 | 0 | 5.55 |
| | $\mathcal{L}_{rec} + \mathcal{L}_{reg}$ | 51.54 | 12.96 | 44.04 | 40.74 | 13.69 | 33.33 |
| | $\mathcal{L}_{rec} + Clip$ | 54.49 | 23.61 | 46.03 | 66.55 | 37.61 | 62.39 |

*Table 13.* Ablation study on TULA-DR against *KL*.

| Model | Setting | SynthPAI-age | | | SynthPAI-inc | | |
|---|---|---|---|---|---|---|---|
| | | R-1 | R-2 | R-L | R-1 | R-2 | R-L |
| Pythia-1.4b | $\mathcal{L}_{rec}$ | 61.11 | 17.5 | 57.4 | 45.83 | 10.31 | 45.83 |
| | $\mathcal{L}_{rec}(l_2\text{-based})$ | 0 | 0 | 0 | 0 | 0 | 0 |
| | $\mathcal{L}_{rec} + \mathcal{L}_{reg}$ | 90.27 | 58.09 | 81.94 | 73.14 | 52.61 | 68.98 |
| | $\mathcal{L}_{rec} + Clip$ | 79.16 | 52.38 | 79.16 | 88.42 | 48.21 | 77.31 |
| OPT-1.3b | $\mathcal{L}_{rec}$ | 7.4 | 0 | 7.4 | 9.39 | 0 | 9.39 |
| | $\mathcal{L}_{rec}(l_2\text{-based})$ | 6.36 | 0 | 6.36 | 9.97 | 0 | 9.97 |
| | $\mathcal{L}_{rec} + \mathcal{L}_{reg}$ | 14.94 | 0 | 11.24 | 9.97 | 0 | 9.97 |
| | $\mathcal{L}_{rec} + Clip$ | 9.52 | 5.55 | 9.52 | 13.42 | 0 | 13.42 |

*Table 14.* Ablation study on TULA-DR against *NPO*.

| Model | Setting | SynthPAI-age | | | SynthPAI-inc | | |
|---|---|---|---|---|---|---|---|
| | | R-1 | R-2 | R-L | R-1 | R-2 | R-L |
| Pythia-1.4b | $\mathcal{L}_{rec}$ | 47.4 | 12.5 | 47.4 | 44.16 | 15.87 | 44.16 |
| | $\mathcal{L}_{rec}(l_2\text{-based})$ | 0 | 0 | 0 | 0 | 0 | 0 |
| | $\mathcal{L}_{rec} + \mathcal{L}_{reg}$ | 63.75 | 41.46 | 56.34 | 55.35 | 23.81 | 55.35 |
| | $\mathcal{L}_{rec} + Clip$ | 88.88 | 71.9 | 88.88 | 70.56 | 38.69 | 70.56 |
| OPT-1.3b | $\mathcal{L}_{rec}$ | 17.18 | 3.33 | 14.15 | 3.7 | 0 | 3.7 |
| | $\mathcal{L}_{rec}(l_2\text{-based})$ | 3.33 | 0 | 3.33 | 9.09 | 0 | 9.09 |
| | $\mathcal{L}_{rec} + \mathcal{L}_{reg}$ | 20.87 | 0 | 17.17 | 35.83 | 9.52 | 35.83 |
| | $\mathcal{L}_{rec} + Clip$ | 60.74 | 31.94 | 49.63 | 61.11 | 28.7 | 50.37 |

*Table 15.* Ablation study on TULA-DR against *TaskVec*.

| Model | Setting | SynthPAI-age | | | SynthPAI-inc | | |
|---|---|---|---|---|---|---|---|
| | | R-1 | R-2 | R-L | R-1 | R-2 | R-L |
| Pythia-1.4b | $\mathcal{L}_{rec}$ | 60.18 | 22.61 | 56.01 | 34.72 | 9.52 | 34.72 |
| | $\mathcal{L}_{rec}(l_2\text{-based})$ | 0 | 0 | 0 | 0 | 0 | 0 |
| | $\mathcal{L}_{rec} + \mathcal{L}_{reg}$ | 67.46 | 24.44 | 57.14 | 79.16 | 44.04 | 71.29 |
| | $\mathcal{L}_{rec} + Clip$ | 91.07 | 68.25 | 86.31 | 53.24 | 36.31 | 53.24 |
| OPT-1.3b | $\mathcal{L}_{rec}$ | 12.03 | 0 | 7.87 | 20 | 3.7 | 20 |
| | $\mathcal{L}_{rec}(l_2\text{-based})$ | 12.5 | 0 | 12.5 | 5.55 | 0 | 5.55 |
| | $\mathcal{L}_{rec} + \mathcal{L}_{reg}$ | 33.59 | 3.7 | 33.59 | 37.56 | 11.11 | 33.86 |
| | $\mathcal{L}_{rec} + Clip$ | 49.93 | 32.93 | 49.93 | 63.78 | 26.66 | 56.37 |

## J. Examples of the Reconstructed Texts via TULA-DR

We list some reconstructed examples in Tab. 16, 17, 18 and Tab. 19, 20, 21, for *SynthPAI-age* and *SynthPAI-inc* datasets respectively. Besides, to fairly evaluate the reconstruction attack, we additionally add padding characters ( for OPT-1.3b, < |endoftext| > for Pythia-1.4b) to fix the length of unlearned texts.

*Table 16.* Reconstructed examples on *SynthPAI-age* dataset.
Reference text: *<padding><padding><padding><padding>strict timelines? more like stress recipes.*

| Model | Method | Reconstructed Texts |
|---|---|---|
| Pythia-1.4b | GA | < \|endoftext\| > nowadays< \|endoftext\| >strict timelines? more like stress |
| | KL | < \|endoftext\| >usi morestrict more timelines? like stress |
| | NPO | < \|endoftext\| >< \|endoftext\| >< \|endoftext\| >< \|endoftext\| >< \|endoftext\| >strictelines. like stress |
| | TaskVec | < \|endoftext\| >< \|endoftext\| >< \|endoftext\| >strict timelines? more like stress |
| OPT-1.3b | GA | like timelines?More likest. stress stressa |
| | KL | like timeframe at plaster Stress stressfula |
| | NPO | like stricter timelines lingurict? Like stressa |
| | TaskVec | like timelines stress coachMore?) timelinesmare Likea |

*Table 17.* Reconstructed examples on *SynthPAI-age* dataset.
Reference text: *<padding><padding>Evolution inevitable; historic sites always layers deep!*

| Model | Method | Reconstructed Texts |
|---|---|---|
| Pythia-1.4b | GA | < \|endoftext\| >< \|endoftext\| >Evolution inevitable; historic sites always layers |
| | KL | Ev< \|endoftext\| >< \|endoftext\| >olution; historic sites layers always layers |
| | NPO | Ev< \|endoftext\| >Evolution sites inevitable; always historic layers |
| | TaskVec | necess< \|endoftext\| >Evolution inevitable; historic sites layers deep |
| OPT-1.3b | GA | likeolution; sites always layers deep horse Hogwartsa |
| | KL | like Ramos Craw fossils favourite aECT ONLY Breakinga |
| | NPO | like;historic sites perenn always just layers deep menstruossusa |
| | TaskVec | likeEvolution inevitable; feudal historic historic sites always layersa |

*Table 18.* Reconstructed examples on *SynthPAI-age* dataset.
Reference text: *<padding>Engineering sure throws curveballs despite hefty spreadsheets!*

| Model | Method | Reconstructed Texts |
|---|---|---|
| Pythia-1.4b | GA | Engineering sure throwsballs despite hefty spreadshe |
| | KL | Engineering spread sure throws curveballs despite hefty |
| | NPO | Engineering sure throwsballs despite hefty spreadshe |
| | TaskVec | Engineering sure throwsballs despite hefty spreadshe |
| OPT-1.3b | GA | likeEngineering sure throws curveballs hearty despite heftysheetsa |
| | KL | like——".######## I toesales lords Tha |
| | NPO | like grotesque tsundespiteborghEngineering sure sly threwsheetsa |
| | TaskVec | likeEngineering throws curveballs despite hefty spreadscffffccsheetsa |

*Table 19.* Reconstructed examples on *SynthPAI-inc* dataset.
Reference text: *<padding><padding>film school grind pays off here - editing gigs underway.*

| Model | Method | Reconstructed Texts |
|---|---|---|
| Pythia-1.4b | GA | pays off here< \|endoftext\| >film school here - editing gig |
| | KL | < \|endoftext\| >< \|endoftext\| >film school pays off here - editing gig |
| | NPO | pays< \|endoftext\| > -< \|endoftext\| >film school here pays editing gig |
| | TaskVec | < \|endoftext\| >film school grind pays off here - editing gig |
| OPT-1.3b | GA | like grind here editing cryptocssl - gigs gigsa |
| | KL | like eagerbreakersknownliesarnaev documentation adversity funds majoritya |
| | NPO | like editing_-_Filmitism Izanft gigs editing pledginga |
| | TaskVec | likefilm school - fishes gigs editing gigs ALECa |

*Table 20.* Reconstructed examples on *SynthPAI-inc* dataset.
Reference text: *<padding>real growth happens inside too - gotta both be evolving*

| Model | Method | Reconstructed Texts |
|---|---|---|
| Pythia-1.4b | GA | realrealFrame growth - both gotta be evolving |
| | KL | real growth - gotta be both - be evolving |
| | NPO | real headed growth happens - gotta be both evolving |
| | TaskVec | real to growth too - both gotta be evolving |
| OPT-1.3b | GA | likereal growth inside shenan egregious - gotta both evolving evolvinga |
| | KL | like growers dope antid Qt activ evolve Damascus Yin Lyme Vermonta |
| | NPO | likereal gotta evolvingrenchesYep Krugutical mysql Okawarua |
| | TaskVec | likereal growth happensburghreal - gotta both be evolvinga |

*Table 21.* Reconstructed examples on *SynthPAI-inc* dataset.
Reference text: *<padding><padding><padding><padding><padding>had tape fix countertop once lol*

| Model | Method | Reconstructed Texts |
|---|---|---|
| Pythia-1.4b | GA | < \|endoftext\| >< \|endoftext\| >had tape fix' countertop tape once |
| | KL | < \|endoftext\| > Veter< \|endoftext\| >had tape fix counter countertop once |
| | NPO | < \|endoftext\| >< \|endoftext\| >< \|endoftext\| >< \|endoftext\| >had tape tape countertop once |
| | TaskVec | < \|endoftext\| >< \|endoftext\| >< \|endoftext\| > countertop fix< \|endoftext\| >had tape once |
| OPT-1.3b | GA | likehadCountertop tape fix Latvia oncea |
| | KL | likeTagsbuf counterOnce tapea |
| | NPO | likeImplhad tape fixtop oncea |
| | TaskVec | likehad tape fixtop tape oncea |

