# OpenReview forum: "Textual Unlearning Gives a False Sense of Unlearning"
_ICML.cc/2025/Conference — ICML 2025 poster_

### Official Review · Reviewer_PnWC · 2025-03-03

**Overall Recommendation:** 2

**Summary:**

The paper investigates the effectiveness of machine unlearning (MU) in LMs and introduces new auditing and attacking methods to evaluate its reliability and privacy risks. They propose U-LiRA+, which uses mislabeled samples to rigorously audit unlearning effectiveness. The results reveal that over 70% of unlearned texts remain detectable. The TULA-MI shows that attackers can infer whether a text was unlearned by comparing model outputs before and after unlearning, even in strict black-box scenarios. The TULA-DR exploits model weight differences to reconstruct unlearned texts with over 80% accuracy in some cases, showing that different unlearning methods leave distinct traces in the model. The findings demonstrate that existing unlearning methods do not ensure true erasure and may even increase privacy risks, highlighting the need for more robust and secure unlearning mechanisms.

**Claims And Evidence:**

The claims are mostly well-supported by empirical evidence. Some claims could benefit from additional validation. However, both TULA-MI and TULA-DR assume that an attacker can access outputs (or even internal weights in the white-box case) from the model before and after unlearning. In practice, such access might be limited, so the generalization of the claims is concerned.

**Essential References Not Discussed:**

No.

**Experimental Designs Or Analyses:**

The choice of using TPR@LowFPR is not justified. Not sure if this metric is used by other work.

**Methods And Evaluation Criteria:**

- The choice of using synthetic data may undermine the claims. I understand the concern that models might have seen the public data. However, the assumptions of this paper are strong (e.g., access to the model weights), which is unlikely to happen in real-world scenarios; it is better to test the model on real-world datasets. There are many other real-world datasets for MU and existing methods to test whether certain data are used to train the model, which could help to construct new datasets if the authors want a pure dataset for unlearning.
- The authors trained 100 shadow models for each shadow example. This can be very time-consuming when their method is extended to commonly used LLMs today. The only imaginable setting I can think of is when a new unlearning method comes out and then it is applied to a smaller model to see if leakage happens under attack. However, even if leakage does not exist, the effectiveness of the new methods remains unknown.

**Other Comments Or Suggestions:**

No.

**Other Strengths And Weaknesses:**

Please refer to the methods section for the weaknesses.

**Questions For Authors:**

Is the choice of Pythia and OPT mainly based on computational efficiency?

**Relation To Broader Scientific Literature:**

This paper challenges prior claims that inexact MU approximates full retraining. They show that unlearning leaves residual traces rather than ensuring true forgetting. The paper contributes to the broader scientific literature by calling for more robust privacy-preserving techniques.

**Theoretical Claims:**

I am not aware of any proof for theoretical claims.

---

> ### Author Rebuttal · Authors · 2025-03-30
>
> # Response to Reviewer PnWC
> We sincerely thank the reviewer PnWC for your valuable and constructive feedback!
> ## Q1: Concerns about our assumptions and their practicality.
> We would like to provide further explanations on our assumptions:
>
> (1) In the **black-box scenario**, we assume that an adversary can *query the model* with target samples and *utilize the outputs to infer their membership*. This is **the most fundamental assumption for Member Inference Attack (MIA)**[1] and is **widely considered in literature as well as real-world applications**. To further align with real-world scenarios, we further investigate **the strictest black-box scenarios**, where the adversary can *only obtain output scores* (e.g., confidence scores) for MIA.
>
> (2) In the **white-box scenario**, we assume that adversary has *access to the model weights before and after unlearning*. This assumption could also be realistic in practice. Here, the adversary could be a **malicious collaborator**, such as *a company contracting with the model developer, with access to model weights for local deployment*. According to the unlearning policy, **the collaborator’s model is also required to be updated after unlearning**, enabling the adversary to have the model weights before and after unlearning. Additionally, this assumption is **commonly studied in literature[2][3]**, serving as **an "upper bound" for exploring real-world risks**.
> ## Q2: Suggestions on utilizing real-world data.
> We utilize synthetic datasets to rigorously ensure that **the pre-trained models have not seen these samples during the pre-training phase**. However, *we are concerned that real-world data can not meet our requirement currently*: (1) Pre-trained models are trained on *vast amounts of real-world data*, which is typically **closed-sourced**. (2) Recent studies indicate that **we can not yet robustly detect whether a specific sample was used among massive pre-training data[4]**.
>
> Therefore, we utilize synthetic samples, which are **entirely based on fictional scenarios**, and generated by **GPT-4-based agents released after the models we utilized**. Through these efforts, we hope to ensure that the setups are rigorous and thus avoid being potentially misleading.
> ## Q3: The concern on computational cost (training multiple shadow models) on the proposed U-LiRA+ for LLM.
> We train multiple shadow models to **ensure the rigorous auditing on unlearning methods**. With them, we could *closely approximate the distributions of the model outputs* and *rigorously determine samples' membership through statistical tests*.
>
> Though this approach could incur much computational cost, **it could also be applied to LLMs**: (1) We could utilize **Parameter-Efficient Fine-Tuning (PEFT)** methods to effectively fine-tune and unlearn few audit samples. Since only few parameters are trainable, the computational cost would be significantly reduced. (2) **Our approach is MIA-agnostic**, as its core idea is to properly construct and inject rigorous audit samples before unlearning. **If more efficient yet equally rigorous MIAs for LLMs emerge, our approach can be flexibly adapted to them.**
> ## Q4: Do we choose the models mainly for computational efficiency?
> **The selection of *1.5B*-parameter models balances *practicality* and *computational efficiency***: (1) **Lightweight LLMs are widely adopted in practice**, such as *autonomous driving* and *mobile AI assistants*. (2) As mentioned, **we aim to ensure rigorous auditing and evaluation**, which requires some consuming operations, e.g., *training multiple shadow models*. Choosing an appropriately sized model enables us to **conduct broader evaluations efficiently**.
>
> Furthermore, we highlight that **the model scale is unlikely to compromise our findings**: (1) We demonstrate that existing auditing methods are not rigorous because *they fail to correctly select audit samples*, **which is independent of the target model**. (2) We reveal that current unlearning methods fail to completely erase target samples because *they cannot accurately identify and remove parameters encoding target knowledge from the large parameter space*, **which would persist across models of different scales.**
> ## Q5: The choice of TPR@LowFPR.
> TPR@LowFPR is the **most commonly used and rigorous metric** for evaluating MIA, quantifying MIA's effectiveness and confidence by measuring **the proportion of correctly inferred samples at a low false rate**[5].
> ## References
> [1] Membership inference attacks against machine learning models, IEEE S&P 2017
>
> [2] Machine Unlearning Enables Camouflaged Poisoning Attacks, NeurIPS 2023
>
> [3] Adversarial Machine Unlearning Requests Destroy Model Accuracy, ICLR 2025
>
> [4] Membership Inference Attacks Cannot Prove that a Model Was Trained On Your Data, IEEE SaTML 2025
>
> [5] Membership Inference Attacks From First Principles, IEEE S&P 2022

---

### Official Review · Reviewer_9rYC · 2025-03-07

**Overall Recommendation:** 4

**Summary:**

This paper critically demonstrates that current machine unlearning mechanisms give a false sense of effective unlearning. First, they propose U-LiRA+, a rigorous textual unlearning auditing method, and find that the unlearned texts can still be detected with very high confidence after the unlearning process. Further, they comprehensively investigate the privacy risks of textual unlearning mechanisms in deployment. By proposing TULA along with its variants in both black- and white-box scenarios, the authors critically reveal that the unlearning mechanism would instead expose more about the unlearned texts, facilitating both membership inference and reconstruction attacks. The experiments demonstrate that the proposed auditing method much more strictly measures the unlearning effectiveness compared to previous approaches. Besides, the proposed TULA-MI and TULA-DR attacks are capable of inferring unlearned samples with high confidence.

Overall, in order to explore the vulnerabilities and privacy risks of machine unlearning in language models, the authors introduce a rigorous auditing method and novel attack paradigms. The experiments support the findings well. In addition, this paper is structured well and easy to follow.

**Claims And Evidence:**

Yes, the claims are clearly supported in the paper.

**Essential References Not Discussed:**

Related works are adequately discussed.

**Experimental Designs Or Analyses:**

Yes, the experimental designs and analyses effectively support the proposed methods.

**Methods And Evaluation Criteria:**

Yes, the proposed methods and evaluation criteria make sense for the problem.

**Other Comments Or Suggestions:**

It would be more informative for the authors to further discuss how your findings inspire the development of unlearning mechanism on language models.

**Other Strengths And Weaknesses:**

NA

**Questions For Authors:**

1. How about the proposed reconstruction attack (TULA-DR) against the exact unlearning method (including retraining) ?
2. How do your findings inspire the development of unlearning mechanism on language models?

**Relation To Broader Scientific Literature:**

1. This work highlights that previous unlearning auditing approaches have overestimated the effectiveness of existing unlearning techniques. The authors propose a novel and rigorous auditing method, U-LiRA+, which could inspire the development of more thorough and reliable unlearning techniques.
2. The work reveals the privacy risks of deploying textual unlearning mechanisms in both black-box and white-box contexts. This serves as a call to further investigate the potential new and additional privacy risks before widespread application of unlearning mechanisms.

**Theoretical Claims:**

There are no theoretical evaluations in this paper.

---

> ### Author Rebuttal · Authors · 2025-03-30
>
> # Response to Reviewer 9rYC
> We sincerely thank the reviewer 9rYC for your valuable and constructive feedback!
>
> ## Q1: How about the proposed reconstruction attack (TULA-DR) against the exact unlearning method (including retraining)?
>
> We mainly focus on TULA-DR against inexact unlearning because: (1) **Exact unlearning is relatively safe**, as its core idea is to delete the target sample and retrain the model from scratch. Due to the randomness in the training process, the difference in model weights before and after unlearning is less likely to be approximated by the adversary. (2) In practical deployment, **developers often use the inexact unlearning methods considering computational efficiency**. Thus, exploring reconstruction attacks against inexact unlearning could provide more practical insights.
>
> ## Q2: How do the findings inspire the development of unlearning mechanism on language models?
>
> Thanks for the insightful question. We hope our findings could inspire future research in the following three aspects:
>
> (1) **Developing more precise unlearning mechanisms**. Considering the difficulty of existing unlearning methods to completely erase target samples, future work could explore *efficient exact unlearning methods* (e.g., retraining) or *certified inexact approaches* capable of accurately identifying target knowledge.
>
> (2) **Strengthening unlearning auditing before real-world deployment**. Future research should establish rigorous auditing frameworks for unlearning mechanisms *from broader perspectives*. Beyond verifying whether an unlearned model successfully erases target data, it is also essential to assess and mitigate potential risks, such as reduced robustness.
>
> (3) **Rethinking the secure deployment of unlearning mechanisms**. While prior research has primarily focused on *how to unlearn*, less attention has been given to *how to deploy unlearning*. As demonstrated in this work, future research should pay more attention to the *secure deployment of unlearning mechanisms* into real-world scenarios while mitigating potential new risks.

---

> > ### Comment · Reviewer_9rYC · 2025-04-03
> >
> > Many thanks for authors' detailed responses. More results and discussions (the method details, insight of algorithms etc.) regarding my concerns and other reviewers have been provided. Based on the overall quality of the paper/response, I’d like to keep my score.

---

> > > ### Author Response · Authors · 2025-04-03
> > >
> > > We sincerely thank the reviewer 9rYC for reviewing our rebuttal, and we appreciate your valuable feedback! We will carefully revise the paper according to your suggestions.

---

### Official Review · Reviewer_tLdQ · 2025-03-10

**Overall Recommendation:** 4

**Summary:**

The authors demonstrate that current unlearning methods fail to adequately protect the privacy of unlearned texts in language models. To address this, they propose a robust unlearning auditing method, U-LiRA+, which utilizes membership inference attacks and deliberately introduces mislabeled samples to reveal that the unlearned texts are, in fact, highly detectable. Furthermore, they introduce textual unlearning leakage attacks, showing that unlearned texts can be inferred within unlearning systems, thereby uncovering a new privacy risk associated with machine unlearning.

**Claims And Evidence:**

The contributions are well supported with sound designs and evaluations.

**Essential References Not Discussed:**

No.

**Experimental Designs Or Analyses:**

The experimental design is sound, and the results empirically demonstrate the effectiveness of the proposed methods.

**Methods And Evaluation Criteria:**

I have some confusions about TULA-DR that need further clarification:(1) How to determine the convergence of TULA-DR? The authors need to further clarify the metrics or criteria for assessing convergence. (2) In Algorithm 3, the authors should add a definition of “Decoding” function.

**Other Comments Or Suggestions:**

No.

**Other Strengths And Weaknesses:**

No.

**Questions For Authors:**

1. How to determine the convergence of TULA-DR? What does the “Decoding” function in Algorithm 3 refer to? While this work focuses on classification tasks, I am curious that could the idea of U-LiRA+ potentially be applied to generation tasks for future works?

2. Although the paper highlights the limitations of text unlearning methods, it lacks an in-depth analysis of the performance differences of existing methods across different data modalities, resulting in its innovation being primarily confined to phenomenological descriptions.

3. While the paper identifies privacy leakage risks in the text unlearning process, it does not employ quantitative analyses based on information entropy or mutual information to demonstrate that the unlearned model retains exploitable information residues. Incorporating information-theoretic approaches, such as computing mutual information between model outputs and the original text or analyzing entropy variations, would provide a more rigorous validation of the inevitability of privacy risks.

4. The paper mentions "high-confidence detection" but does not explicitly define a statistical significance threshold, which may affect the robustness of its conclusions. For instance, failing to adopt a stringent significance level in hypothesis testing could lead to an increased false positive rate.

**Relation To Broader Scientific Literature:**

The authors rethink the security of unlearning mechanisms on language models, revealing the vulnerabilities of existing methods from auditing to deployment.

**Theoretical Claims:**

The authors did not present any theoretical proofs.

---

> ### Author Rebuttal · Authors · 2025-03-30
>
> # Response to Reviewer tLdQ
> We sincerely thank the reviewer tLdQ for your valuable and constructive feedback!
>
> ## Q1: How to determine the convergence of TULA-DR?
> Our proposed TULA-DR is an **optimization-based attack**. Empirically, the optimized candidates converge gradually as the number of iterations increases. A criterion is that the loss of the attack **stabilizes at a lower value and no longer continues to decrease**. Therefore, when implementing the attack, we could **set a maximum number of iterations to stop the optimization process**.
>
>
> ## Q2: What does the “Decoding” function in Algorithm 3 refer to?
>
> This function refers to transforming the reconstructed embeddings into **text space**. To implement this function, we use the model’s **embedding layer** as a *“vocabulary”* and apply a similarity computation function (e.g., *cosine similarity*) to **match the ordinal indices of the reconstructed embeddings**. The resulting list of indices is then converted into readable text using a **tokenizer**.
>
> ## Q3: Can U-LiRA+ potentially be applied to generation tasks for future work?
>
> Our approach is **task-agnostic**, as its core idea is to properly construct and inject rigorous audit samples before auditing. Specifically, in order to implement rigorous unlearning auditing on a text generation task, we should first **define the samples that are most vulnerable to unlearning on that task as audit samples**. We then **inject these audit samples into the training set** and evaluate whether the target unlearning method could fully erase them. In addition, to implement U-LiRA+ on the generation task, we could **utilize the next-word probability vector in response to a text sample as the model's *output* for MIA**.
>
> ## Q4: In-depth analysis on the performance differences of existing methods across different data modalities.
> This study focuses on textual data and language models. However, our key findings could be broadly applicable to various data modalities:
>
> (1) Existing unlearning methods can not fully erase target samples, primarily due to **their inability to accurately identify parameters linked to target knowledge**. This limitation could extend to other data modalities.
>
> (2) Analyzing models before and after unlearning allows an adversary to infer membership information about unlearned samples or even reconstruct them. **While different data modalities may exhibit varying degrees of privacy leakage, the underlying risk persists**.
>
> ## Q5: The suggestion for quantitatively analyzing the information residuals of unlearned models.
> We thank you for your insightful suggestion, and it is indeed a very interesting perspective.
>
> However, our proposed auditing method is currently sufficient to demonstrate that **existing unlearning methods can not completely erase the target samples and result in large information residuals**. With these findings, we hope to call for the development of more accurate unlearning methods before applying them in practice. Indeed, with the development of unlearning techniques, how to effectively and exactly measure **“how much”** information residual with theoretically guarantee will be an important problem. We thank you for your inspiration and would be happy to explore this question in future research.
>
> ## Q6: The concern on "high-confidence detection" of our proposed U-LiRA+.
> The reason for our claim is that we utilize the TPR@LowFPR metric. TPR@LowFPR is the most commonly used and rigorous metric for evaluating MIA[1], as it quantifies an attack's effectiveness and **confidence** by measuring **the proportion of correctly inferred samples at a low false rate**.
> ## References
> [1] Membership Inference Attacks From First Principles, IEEE S&P 2022

---

> > ### Comment · Reviewer_tLdQ · 2025-04-03
> >
> > All my concerns have been addressed, so I recommend accepting it.

---

> > > ### Author Response · Authors · 2025-04-03
> > >
> > > We sincerely thank the reviewer tLdQ for reviewing our rebuttal, and we appreciate your valuable feedback! We will carefully revise the paper according to your suggestions.

---

### Official Review · Reviewer_c3T7 · 2025-03-12

**Overall Recommendation:** 4

**Summary:**

The authors demonstrate that the textual unlearning mechanism can not ensure privacy as expected. They propose a rigorous unlearning auditing method (U-LiRA+) and and investigate privacy attacks in both black-box and white-box scenarios. Through empirical evaluations on large language models and synthetic datasets, the authors reveal that existing textual unlearning methods fail to completely erase target texts. Furthermore, these methods may inadvertently expose additional information about unlearned texts through membership inference attacks (MIA) or data reconstruction attacks (DRA).

**Claims And Evidence:**

The claims are well-supported.

**Essential References Not Discussed:**

No

**Experimental Designs Or Analyses:**

The experimental evaluations effectively validate the performance of the proposed auditing method and the attacks in the paper.

**Methods And Evaluation Criteria:**

The proposed methods and evaluations are sound, but there are several minor issues.
1. I am confused about how to initialize the candidates for the proposed TULA-DR.
2. It would be better to try more metrics to evaluate the difference between loss values to implement the proposed MIA.

**Other Comments Or Suggestions:**

There are some suggestions:
1) Algorithm 2 appears crowded and the authors could improve its clarity;
2) It would be helpful if the authors could color-code the reconstructed texts in Tables 16-21 to better differentiate correct and incorrect results;
3) It appears that Fig.3 misses the “%”.
4) The introduction of U-LiRA+ is compelling, but consider adding a brief explanation of why mislabeled samples represent a "worst-case scenario" for auditing. This would improve the clarity.
5) The paper lacks a description on compute requirements (e.g., GPU, CUDA).

**Other Strengths And Weaknesses:**

Weaknesses：
1. Considering that LiRA may be time-consuming in some cases, is it possible for the proposed U-LiRA+ to be adopted to other MIAs?
2. For the TULA-DR, how do the authors initialize the candidates?
3. For the TULA-MI in strict black-box case, the authors utilize the difference in loss values to implement the proposed MIA. Considering that the loss changes may be non-linear, what about other ways of calculating the difference?

**Questions For Authors:**

N/A.

**Relation To Broader Scientific Literature:**

This work shows that even in the strict black-box querying and exact unlearning scenario, the unlearning mechanism can still compromise the privacy of unlearned data. This provides new insights into the secure deployment of unlearning mechanisms.

**Theoretical Claims:**

The work does not include any theoretical proofs.

---

> ### Author Rebuttal · Authors · 2025-03-30
>
> # Response to Reviewer c3T7
> We sincerely thank the reviewer c3T7 for your valuable and constructive feedback!
> ## Q1: Is it possible for the proposed U-LiRA+ to be adopted to other MIAs?
> Yes, our approach is **MIA-agnostic** (membership inference attack), as its core idea is to **properly construct and inject rigorous audit samples before auditing**. For an unlearning auditing task, we start by defining a set of **the most vulnerable samples** (e.g., mislabeled samples) as **audit samples** on that task. Use half of them as training samples and half as unseen samples. After unlearning the training samples with the target unlearning method, we can utilize **any kind of MIA** to test whether it is possible to successfully distinguish between the training samples and the unseen samples in the audit set.
>
> ## Q2: For the TULA-DR, how do the authors initialize the candidates?
> We randomly initialized the candidate embeddings with Gaussian distribution. Additionally, it is also acceptable to utilize other approaches for initialization, such as a random-selected sentence's embeddings or uniform distribution. Empirically, the key insight is that *the values of the initialized embeddings are preferably distributed within the value domain of normal embeddings*.
>
> ## Q3: What about other ways of calculating the difference for the TULA-MI in the strict black-box case?
> Thank you for this insightful question. Indeed, we used the value difference (i.e., A-B) to capture the loss changes on the model before and after unlearning the target sample. However, **other approaches such as ratio and logarithm are also acceptable**, as long as there is eventually a value to quantify the loss change.
>
> ## Q4: Why do mislabeled samples represent a "worst-case scenario" for auditing?
> Compared to normal samples, **mislabeled samples** are a small number of counterfactual samples, ensuring the model cannot generalize to them unless explicitly trained. As a result, the trained and unseen **mislabeled samples** will be very different in output distributions and thus are the *most vulnerable to unlearning auditing*. Therefore, we inject the **mislabeled samples** into the training set as the audit samples, in order to simulate the worst-case auditing.

---

### Official Review · Reviewer_p1Cm · 2025-03-13

**Overall Recommendation:** 4

**Summary:**

The paper proposes a new auditing method to check whether unlearning text from a model is completely unlearned. The auditing method called U-LiRA+ is based on U-LiRA and checks whether it is possible to differentiate between unlearned and not seen samples. Additionally two methods for investigating privacy risks for unlearned models are presented called TULA-MI and TULA-DR. TULA-MI is testing whether an adversary can tell that a specific datapoint was unlearned while TULA-DR tries to reconstruc the data point after unlearning.

**Claims And Evidence:**

The claims are empirically supported. However, the presentation of the experimental results could be clearer (see comment regarding Figure 2).

**Essential References Not Discussed:**

- I am not quite sure what the difference between TULA and the work of Chen et al. [1] is. This should be thoroughly discussed, as they also have proposed a membership inference attack that is able to tell whether a data point was unlearned or not.
- It should be made clearer what the difference between U-LiRA+ and U-LiRA from Hayes et al. [2] is.

[1] Chen et al., When machine unlearning jeopardizes privacy, CCS 2021
[2] Hayes et al, Inexact unlearning needs more careful evaluations to avoid a false sense of privacy, preprint 2024

**Ethical Review Concerns:**

There are no ethical reviews needed for this paper.

**Experimental Designs Or Analyses:**

The experimental design seems to be sound.

**Methods And Evaluation Criteria:**

The experimental design seems to make sense, even though the proposed approaches are only tested on two synthetic datasets.

**Other Comments Or Suggestions:**

- Line 143 right side: there is a point after "M_original" that should probably not be there
- it is not possible to highlight any text in the pdf with the mouse or click on any links. Please fix!
- Line 304 left side: two times "However,"
- Line 320 right side: "fine-turned" should be fine-tuned
- Figure 2: in the legend for the blue dotted line it says "Audit M_original", but I think this should be "Audit M_unlearned". At least that is what the caption suggests.
- Line 412 right side: R-3 is used, but this metric is not used in the table

**Other Strengths And Weaknesses:**

Weaknesses:
- in my opinion the argument in TULA that users have access to the models before and after unlearning is a bit unrealistic to me. After all the users don't know when the model will be updated and they don't have access to the model weights either.
- it is a bit confusing that the paper talks about language models, but in the end only looks at sentiment analysis models aka. models that classify input text.

**Questions For Authors:**

1. It is not clear to me what the dashes lines in Figure 2 are.  What does audit on the original and the unlearned model mean? Why is there only a single blue line, even if there are multiple unlearned models?
2. What is the metric "NTS@1NFS"? This should be explained or written out at least once.
3. Why are there so many "</s>" before the synthetic texts?

**Relation To Broader Scientific Literature:**

The findings are not really surprising. Previous work (Hayes et al.) has already shown that unlearning on text needs better evaluation methods.

**Theoretical Claims:**

There are no theoretical claims.

---

> ### Author Rebuttal · Authors · 2025-03-30
>
> # Response to Reviewer p1Cm
> We sincerely thank the reviewer p1Cm for the valuable and constructive feedback!
> ## Q1: Clear the differences between the proposed TULA and [1].
> Here are the key differences:
>
> (1) **Broader and more realistic assumptions.** ***[1] considers only the relaxed black-box scenario***, where the adversary performs membership inference attacks (MIA) with access to output logits, model architecture and auxiliary dataset. In contrast, ***our proposed TULA could be further applied to a strict black-box scenario***, where the adversary can only obtain output scores (e.g., confidence values), ***providing more practical insights on real-world risks.***
>
> (2) **More powerful attack paradigm.** For the target sample *x*, [1] trains the attack model using logits from ***randomly sampled unseen samples as negative supervision***, resulting in a ***coarse-grained, average-level MIA***. In contrast, we ***train multiple shadow models to accurately learn the logits when *x* is unlearned or unseen***, enabling ***fine-grained, sample-level MIA***.
>
> (3) **Target beyond MIA.** While [1] focuses solely on MIA, we propose both ***MIA*** and ***data reconstruction attack*** against unlearning mechanisms, exploring the risks from ***adversaries with different targets***.
>
> (4) **Focus on more challenging setting**. [1] focus only on tabular and image data with ML models or CNN—***traditional setups where MIAs are much easier due to high distinctiveness among samples***. However, ***MIAs on textual data are much more challenging, necessitating a stronger attack*** [3]. Our results show that [1] is much ineffective against textual data and modern language models.
> ## Q2: Clear the differences between the proposed U-LiRA+ and [2].
> Compared to [2], we **explicitly construct and inject a rigorous audit set with *mislabeled samples* before unlearning, ensuring a rigorous auditing in the worst case**. We demonstrate that previous auditing methods, including [2], **lack rigor due to their failure to properly select audited data**, resulting in **existing unlearning methods being significantly overestimated**.
> ## Q3: How does the adversary know when unlearning has occurred, and have access to model weights?
> (1) **The occurrence of unlearning could be easily detected in practice**. **An adversary can *continuously* query the model with target samples**. If the outputs from two consecutive sets of queries **change**, *the adversary could execute the attack to infer whether any samples have been unlearned*.
>
> (2) **The access to model weights could also be realistic**. The adversary could be a **malicious collaborator**, such as *a company contracting with the model developer, with permission to model weights for local deployment*. According to the unlearning policy, **the collaborator’s model is also required to be updated after unlearning**, enabling the access to model weights before and after unlearning. Moreover, *this assumption is widely considered in literature*[4], serving as **an "upper bound" for exploring real-world security risks**.
> ## Q4: Why do we focus on text classification tasks?
> **Text classification tasks enabling us to design *rigorous* audit samples for *rigorous* unlearning auditing**. Specifically, **rigorous auditing requires rigorous audit samples, i.e., *most vulnerable samples* to unlearning, enabling the worst-case auditing**. The utilized ***mislabeled samples***  have found to be most vulnerable to privacy leakage on image classification tasks[5]. Although our approach is **task-agnostic**, **rigorous samples for text generation tasks are currently under-explored**.
> ## Q5: The Confusions on Figure 2.
> We would like to explain Figure 2 based on the process of unlearning auditing: The audit set consists of training and unseen samples. For **one original model**, **multiple unlearned models** are derived by applying *different unlearning methods* to the training samples in the audit set. MIA-based auditing methods are expected to exhibit high attack accuracy on the **original model** (**dashed lines in Fig. 2**) and low accuracy on **unlearned models** (**bars in Fig. 2**).
> ## Q6: The Meaning of Metric "NTS@1NFS".
> NTS@1NFS refers to *the Number of True Samples @ 1 False Sample*, quantifying **the number of correctly inferred samples with only one error** for MIAs.
> ## Q7: The occurrence of "<\/s>" before the synthetic texts.
> To fairly evaluate the reconstruction attack, we additionally add padding characters (<\/s>) to fix the length of unlearned texts.
> ## References
> [1] When Machine Unlearning Jeopardizes Privacy, CCS 2021
>
> [2] Inexact Unlearning Needs More Careful Evaluations to Avoid a False Sense of Privacy, preprint 2024
>
> [3] Do Membership Inference Attacks Work on Large Language Models?, COLM 2024
>
> [4] Adversarial Machine Unlearning Requests Destroy Model Accuracy, ICLR 2025
>
> [5] Evaluations of Machine Learning Privacy Defenses are Misleading, CCS 2024

---

> > ### Comment · Reviewer_p1Cm · 2025-04-02
> >
> > Thank you very much for the clarifications.
> >
> > **Q1/Q2:** Thank you for clarifying the novelty. I can see now that TULA is different from previous works.
> > **Q3:** Thank you for clarifying the setting. Yes, indeed, these assumptions are realistic for an upper bound.
> >
> > **Q4:** Thank you for the clarification. You say that TULA-MI is task agnostic. However, I don't see how you can apply that approach to a generative task. While for text classification, you have several tokens as input and a single logit vector is output, with a generation task, there are multiple logit vectors for a single sample (one logit vector for each token).
> >
> > **Could you clarify how TULA is task-agnostic in this case?**
> >
> > **Q5:** Thank you for the detailed description. This description should be added to the figure. Additionally, I think it would make sense to edit the figure so that the values do not overlap and all the values can be read. However, I am still not sure what exactly is meant by "Baseline."
> >
> > **Which approach are you comparing to here?**
> >
> > **Q6:** Thank you for clarifying the metric. **Could you explain why you chose this metric instead of sticking with the TPR@x%FPR metric?** If I am not mistaken, the meaning/expressiveness of both metrics are the same.
> >
> > **Q7:** Thank you for clarifying.

---

> > > ### Author Response · Authors · 2025-04-02
> > >
> > > # Further Response to Reviewer p1Cm
> > > We sincerely thank the reviewer p1Cm for reviewing our rebuttal, and we appreciate your valuable feedback!
> > >
> > > Below is a further explanation of your latest feedback:
> > >
> > > ## Q4: Why is the TULA task-agnostic?
> > > For generative tasks, we can use **the predicted next-word logit vector** of a text sample as its **output**, a widely adopted approach in existing MIAs for generative tasks[1][2]. The next-word logit vector effectively indicates whether the model has been trained on the target text. For instance, the next-word prediction distribution may **converge to a specific word** for a *training sample* or **diverge** for an *unseen sample*.
> > >
> > > Thus, for generative tasks, TULA-MI can be implemented using the next-word logit vector of the target sample. In other words, **for both classification and generative tasks, the input sample always produces one logit vector output for conducting MIA, making our approach task-agnostic**.
> > >
> > > ## Q5: Which approach are we comparing in Figure 2？
> > > We compare our proposed auditing method with **U-LiRA[3]**, a previously regarded rigorous auditing method. Our results indicate that *the method significantly overestimates the effectiveness of existing unlearning methods*.
> > >
> > > **We will carefully revise Figure 2 as you suggested**.
> > >
> > > ## Q6: Why do we utilize the NTS@1NFS?
> > > **NTS@1NFS represents the strictest case for TPR@x%FPR**, measuring how many correct samples an adversary can confidently infer **while allowing only a single mistake**. To rigorously evaluate TULA-MI *in the strict black-box scenario* (Section 4.2.1), we apply such a strict metric to **highlight the worst-case privacy leakage** caused by *a highly cautious adversary*.
> > >
> > >
> > > ## References
> > > [1] Extracting Training Data from Large Language Models, USENIX Security 2021
> > >
> > > [2] Membership Inference Attacks against Language Models via Neighbourhood Comparison, ACL 2023
> > >
> > > [3] Inexact Unlearning Needs More Careful Evaluations to Avoid a False Sense of Privacy, preprint 2024

---

### Decision · Program_Chairs · 2025-05-01

**Decision:**

Accept (poster)

**Comment:**

This paper proposes a rigorous auditing method for unlearning to demonstrate that current textual unlearning mechanisms cannot guarantee privacy as expected. In addition, two methods, i.e., TULA-MI and TULA-DR, are introduced to investigate the privacy risks of unlearned models. However, using synthetic data in the experiments may raise concerns about the practicality of the proposed approach. After reading the authors' response, I believe they can clearly explain the reason for using synthetic samples based on fictional scenarios in the camera-ready version. Therefore, I recommend acceptance of the paper.